

**Core and margin in warm convective clouds. Part II: aerosol effects**
**on core properties**
[1]Reuven H. Heiblum, [1]Lital Pinto, [1]Orit Altaratz, [1]Guy Dagan, [1]Ilan Koren
[1]Department of Earth and Planetary Sciences, Weizmann Institute of Science, Rehovot, Israel
Corresponding Email – ilan.koren@weizmann.ac.il



**Abstract:**
The effects of aerosol on warm convective cloud cores are evaluated using single
cloud and cloud field simulations. As presented in Part I, the $B_{core} \subseteq RH_{core} \subseteq W_{core}$
property is seen during growth of warm convective clouds. We show that this
property is kept irrespective of aerosol concentration. During dissipation core
fractions generally decrease with less overlap between cores. However, for clouds that
develop in low aerosol concentrations capable of producing precipitation, $B_{core}$ and
subsequently $W_{core}$ volume fractions may increase during dissipation (i.e. loss of
cloud mass). The $RH_{core}$ volume fraction decreases during cloud lifetime and shows
minor sensitivity to aerosol concentration.
It is shown that a $B_{core}$ forms due to two processes: i) Convection – condensation
within supersaturated updrafts and release of latent heat, ii) Adiabatic heating due to
weak downdrafts. The former process occurs during cloud growth for all aerosol
concentrations. The latter process only occurs for low aerosol concentrations during
dissipation and precipitation stages where large mean drop sizes permit slow
evaporation rates.
The aerosol effect on the diffusion efficiencies play a crucial role in the development
of the cloud and its partition to core and margin. Using the $RH_{core}$ definition, it is
shown that the total cloud mass is mostly dictated by core processes, while the total
cloud volume is mostly dictated by margin processes. Increase in aerosol
concentration increases the core (mass and volume) due to enhanced condensation but
also decreases the margin due to evaporation. In clean clouds larger droplets
evaporate much slower, enabling preservation of cloud volume and even increase by
dilution (detrainment while losing mass). This explains how despite having smaller
cores and less mass, cleaner clouds may live longer and grow to larger sizes.







## 1. Introduction

Aerosols remain one of the largest sources of uncertainty in climate predictions, mainly via their effects on clouds (IPCC, 2013). Here we focus on the aerosol effects on warm clouds. Aerosols act as cloud condensation nuclei (CCN) during heterogeneous nucleation by reducing the supersaturation required for droplet activation (Köhler, 1936;Mason and Chien, 1962), yielding differences in the initial cloud droplet size distribution (DSD). Polluted clouds have more, but smaller droplets, and a narrower DSD compared to clean clouds (Twomey, 1977;Andreae et al., 2004). Changes in the initial DSD drive various effects and feedbacks on the cloud's evolution and key processes, such as: droplet mobility, condensation/evaporation budgets, collision-coalescence, and entrainment (Jiang et al., 2006;Xue and Feingold, 2006;Small et al., 2009;Koren et al., 2015).

It is well known that an abundance of small droplets in a cloud (a narrow DSD) reduces the efficiency of the collision-coalescence process (Squires, 1958;Warner, 1968;Twomey, 1977), prolongs the diffusional growth time (Khain et al., 2005;Wang, 2005), and delays or even completely suppresses the initiation of precipitation (Hudson and Yum, 2001;Hudson and Mishra, 2007;L'Ecuyer et al., 2009;Albrecht, 1989). Moreover, in-cloud condensational growth is more efficient in consuming supersaturation because of the larger surface area-to-volume ratio of droplets (Mordy, 1959;Reutter et al., 2009;Pinsky et al., 2012;Seiki and Nakajima, 2013;Dagan et al., 2015a, b). These processes above enable the more polluted cloud to condense more water and intensify its growth via increased release of latent heat (Kogan and Martin, 1994;Koren et al., 2014). The smaller droplets are also pushed higher in the atmosphere due to larger droplet mobility (Koren et al., 2014;Koren et al., 2015)).

However, the increase in aerosol amount yields suppressing effects as well. The symmetry of the diffusion equation dictates that an opposite effect should take place in the sub saturated regions of the cloud, where more numerous and smaller droplets increase the evaporation rate and loss of cloud mass. Increased evaporation can promote entrainment mixing which in turn mixes more sub saturated air into the cloud and further promotes evaporation (Jiang et al., 2006;Xue and Feingold, 2006;Small et al., 2009), effectively initiating a positive feedback between evaporation and mixing with the eventual suppression of cloud growth. This effect may also be accompanied



by a suppressing effect of the larger water loading in polluted clouds which contain
more liquid water mass.
The competition between those opposing processes that are driven by enhanced
aerosol loading determines the net aerosol effect on cloud properties such as cloud
fraction, lifetime, albedo, mass, size, and precipitation amount. However, the sign and
magnitude of such effects are non-trivial (Jiang and Feingold, 2006;Stevens and
Feingold, 2009). Previous studies report opposing findings regarding the total aerosol
effects on warm clouds (Altaratz et al., 2014). Some studies suggest cloud
invigoration by aerosols (bigger and deeper clouds) (Kaufman et al., 2005;Dey et al.,
2011;Yuan et al., 2011;Koren et al., 2014) while some suggest cloud suppression or
no effect at all (Jiang and Feingold, 2006;Xue et al., 2008;Li et al., 2011;Savane et al.,
2015). Moreover, other work has shown that the precipitation susceptibility (i.e.
quantifies the sensitivity of precipitation to the aerosol increase) has a non-monotonic
behavior that reaches its maximum at intermediate LWP values (Sorooshian et al.,
2009), implying that the resultant aerosol effects are heavily dependent on cloud type
and environmental conditions.
A unified theory for the contradicting results regarding aerosol effects was shown in
recent work (Dagan et al., 2015b).  It was shown that the competition between
opposite processes leads to an optimum value of aerosol concentration regarding
various cloud properties like total mass, cloud top, or rain. A cloud that develops
under low aerosol concentration is aerosol limited, as it does not have enough
collective droplet surface area to consume the available water vapor. On the other side
of the trend, a cloud that develops in polluted environment (with more aerosols than
the optimum) is influenced significantly by enhanced entrainment and larger water
loading, causing suppression of cloud development. The optimal concentration is a
function of the thermodynamic conditions (temperature and humidity profiles) and
cloud size.
Environments that support larger clouds development will have larger cloud cores that
are positively affected by aerosol increase and can be regarded as aerosol limited (i.e.
on the ascending branch of the aerosol trend) up to a higher optimal aerosol
concentration. Environmental conditions that support small clouds are more strongly
affected by cloud suppression processes at the cloud margins (due to higher cloud


surface area to volume ratio) and would have a lower optimal aerosol concentration.
This can explain why studies biased to smaller clouds (mostly numerical modeling
studies) report cloud suppression and studies biased to larger clouds (mostly
observational studies) report cloud invigoration. Similar conclusions were reached for
the cloud field scale as well (Dagan et al., 2017).
In addition, it was shown that clouds impact differently the environmental
thermodynamics according to the aerosol level in the field (Seifert and Heus,
2013;Seifert et al., 2015;Dagan et al., 2016). For example changes in aerosol loading
impact the amount of precipitation reaching the surface and subsequently the
evaporative cooling below cloud base and the organization patterns (Xue et al.,
2008;Seifert and Heus, 2013;Seigel, 2014). Moreover, an increase in aerosol loading
may increase evaporation rates around the margins and tops of clouds (Xue and
Feingold, 2006;Stevens, 2007;Seigel, 2014), cooling the upper cloudy layer and
increasing the convective instability. Therefore aerosol effects on phase changes and
precipitation result in vertical redistribution of heat and moisture, which may either
stabilize or destabilize the environment in which subsequent clouds grow (Seifert and
Heus, 2013).
Irrespective of the definition chosen, the cloud's core and margin are dominated by
different processes (Dagan et al., 2015b). These processes often compete with each
other, with the dominant one changing along the cloud's evolution. For example, at
the initial stage of cloud formation, a cloud is more adiabatic and is controlled by the
core's processes (condensation), and when it dissipates the margin processes are more
dominant (entrainment and evaporation). Aerosols affect each of these processes and
thus each stage in the cloud's lifetime.

**2. Methods**
The analyses performed here are to the most part identical to those described in Part I
(hereafter PTI) of this work. In this section we shall thus only give a brief review of
the methods used. For single cloud simulations we use the Tel-Aviv University
axisymmetric cloud model (TAU-CM (Reisin et al., 1996)), and for cloud field
simulations we use the System for Atmospheric Modeling (SAM) Model (version



6.10.3, for details see webpage: http://rossby.msrc.sunysb.edu/~marat/SAM.html)
(Khairoutdinov and Randall, 2003)).
Both models utilize explicit bin microphysics schemes (Tzivion et al., 1987;Khain et
al., 2004), solving nucleation, diffusion (i.e. condensation and evaporation),
collisional coalescence, breakup, and sedimentation microphysical processes. The
single cloud model is initialized using a Hawaiian thermodynamic profile, based on
the 91285 PHTO Hilo radiosonde at 00Z, 21 Aug, 2007. The cloud field model is
setup based on the BOMEX case study, including an initialization setup (sounding,
surface fluxes, and surface roughness) and large scale forcing setup (Siebesma et al.,
2003). More details on the model setups and definitions can be found in PTI.
To study the effects of aerosols on the cloud cores we run each model setup with three
different aerosol concentrations: clean – 25 cm$^{-3}$, intermediate – 250 cm$^{-3}$, and
polluted – 2000 cm$^{-3}$. As defined in Part I, all pixels with at least 0.01 g kg$^{-1}$ of liquid
water are considered cloudy. Cloud cores are defined using three definitions:  1)
$RH_{core}$: relative humidity > 100%, 2) $B_{core}$: buoyancy > 0, and 3) $W_{core}$: vertical
velocity > 0. Relative humidity (RH) and vertical velocity (W) are standard outputs of
the model, while the buoyancy (B) is calculated based on eq. 1 in PTI, where each
cloudy pixel is compared with the mean non-cloudy thermodynamic reference state
per height.
In order to reduce the problem's dimensionality and distill signals in a cloud field
system governed by high variance, we use the Gravity vs. Mass (CvM) phase space in
combination with an automated 3D cloud tracking algorithm (Heiblum et al., 2016b).
The CvM phase space enables a compact view of all clouds in the simulation, by
projecting only their Center-of-Gravity (COG) height and mass at each output time
step. Using the cloud tracking, it was shown that the lifetime of a cloud can be
described by a trajectory on this phase space. Hence, the different locations in the
CvM space are associated with different stages in a cloud's lifetime (i.e. growing,
precipitating, and dissipating). For an in-depth explanation of the CvM space, the
reader is referred to Sect. 2.4 in PTI (see schematic illustration - Fig. 1, PTI).




### 3. Results – Single cloud simulations

#### 3.1. Sensitivity of different core types to aerosol concentration

Figure 1 presents time series of single cloud core volume fractions and cores' properties, for three aerosol concentrations (clean, intermediate, and polluted). Also included are time series of instantaneous rain-rates [mm hr$^{-1}$] at the domain surface. For all aerosol concentrations and during most of the clouds' lifetimes, the volume fraction of $W_{core}$ tends to be the largest and of $B_{core}$ the smallest. Exceptions to this finding are seen either at the initial time step for the polluted cloud or the later stages of cloud lifetime for the lower concentration clouds. In addition, we find that $RH_{core} \subseteq W_{core}$ for all stages of cloud lifetime while $B_{core} \subseteq W_{core}, RH_{core}$ for all stages of the polluted cloud but only applies to the growing stages of lower concentration clouds before precipitation production. Thus, the main finding from PTI (i.e. $B_{core} \subseteq RH_{core} \subseteq W_{core}$) generally applies to all aerosol concentrations during the pre-precipitation stages of the clouds' lifetimes.

Lower aerosol concentration simulations produce more rain, and at earlier stages of cloud lifetime due to efficient collision coalescence. The increase in $B_{core}$ volume fraction at later stages of cloud lifetime in those simulations (clean and intermediate) coincides with initiation of precipitation production, followed by a consequent increase in $W_{core}$ volume fraction as well (more so for the intermediate concentration). The possible mechanism behind the increase in prevalence of buoyant parcels during precipitation is explored in Sect. 3.2. The lack of $RH_{core}$ pixels at these stages indicates that the $W_{core}$ is composed of pixels with small vertical velocities, insufficient for supersaturation production. The $RH_{core}$ is the only one which is not sensitive to rain and monotonically decreases during all clouds' lifetimes. Another clear aerosol effect seen in Fig. 1 is an increase in cloud lifetime with decrease in aerosol concentration. This point will be further explored in Sect. 3.3.

#### 3.2. Mechanisms governing positive buoyancy

The theoretical arguments in PTI showed that $B_{core}$ should be the smallest of the three. This was shown for both the adiabatic cloud column case and also the non-adiabatic case where entrainment mixing and consequent evaporation has a strong net



negative effect has on cloud buoyancy. Despite this fact, results show (see Fig. 1, and
Fig. 2 in PTI) that pockets of positive buoyancy may form independent of the other
cores during dissipation and precipitation stages, even though evaporation is to be
expected then. Since positive buoyancy is the result of either higher temperature or
vapor content (or both) than the surrounding environment, we choose to analyze these
two terms during different stages of the single cloud lifetimes.
Figure 2 shows the values of the temperature ($B_T$) and humidity ($B_{Qv}$) buoyancy terms
in pixel buoyancy vs. pixel vertical velocity phase space. The scatter plots include all
cloudy pixels during all time steps, for the three different aerosol concentration
simulations. The distribution of points for the polluted simulation shows a positive
linear dependence of buoyancy on vertical velocity. Negative vertical velocity is
associated with negative buoyancy and positive vertical velocity shows a transition
from negative to positive buoyancy with increase in magnitude. For this case both $B_T$
and $B_{Qv}$ increase with increase in vertical velocity, as is generally expected in
convective clouds. The sign of pixel buoyancy is mostly dependent on $B_T$ since all
pixels have positive $B_{Qv}$ and a negative water loading term. This behavior is also seen
for lower aerosol concentrations, where the sign of buoyancy is to the most part
determined by $B_T$.
The clean and intermediate simulations show a similar dependence of buoyancy on
vertical velocity; however, it is apparent that these simulations also include an outlier
scatter region of pixels with positive buoyancy and weak negative vertical velocity
which is absent in the polluted simulation (see white arrows, Fig. 2). Consistent with
the rest of the cloudy pixels, these outlier pixels have positive $B_T$, but differ in that
they show neutral $B_{Qv}$. It can also be seen that these pixels are only attributed to the
stages after surface precipitation has commenced (indicated by black dots in markers).
Precipitation is indicative of both downdraft motion and abundance of large droplet
sizes.
Thus, we hypothesize that pockets of positive buoyancy may form due to transport of
parcels with higher potential temperature from above, namely adiabatic heating. The
weak downdrafts also transport lower mixing ratio ($Q_v$) values, as is indicated by the
neutral $B_{Qv}$. Moreover, the occurrence during precipitating stages and for lower
aerosol concentrations indicates that slow evaporation due to larger droplet sizes is



crucial. Indeed, most pixels with negative buoyancy show positive $B_{Qv}$ except for the
clean case where rain pixels from the cloudy layer sediment well below the cloud base
and experience higher environmental Qv (while evaporating slowly), resulting in
negative $B_{Qv}$.

**3.3.    The dependency of cloud characteristics on core and margin's**
**processes**
Here we evaluate how aerosol effects within the core and margin affect the cloud
characteristics, focusing on two main parameters; size (or volume) and mass. In Fig. 3
we follow the evolution of cloud, core, and margin mass and volume for different
aerosol concentrations, using only the $RH_{core}$ definition. We choose the $RH_{core}$ since
it is the most well behaved out the core types, generally decreasing monotonically
(see Fig. 1). A non-monotonic dependency of total cloud mass on aerosol
concentration is seen, showing a maximum for the intermediate concentration. This
type of dependency has been previously reported for warm cumulus clouds (Dagan et
al., 2015b;Savane et al., 2015).
One can generally expect an increase in diffusion and decrease in collision-
coalescence processes efficiency with increase in aerosol concentration (Hudson and
Yum, 2001;Jiang et al., 2009;L'Ecuyer et al., 2009;Pinsky et al., 2012), affecting both
condensation and evaporation processes. The intermediate concentration shows the
highest total mass as a result of being an optimal case with higher condensation
efficiency than the clean case and lower evaporation efficiency than the polluted case.
It is convenient to represent the condensation and evaporation efficiencies by the
$RH_{core}$ and $RH_{margin}$ mass, respectively. The intermediate cloud has almost identical
core mass as does the polluted cloud, but retains higher mass in its margin as well.
The clean cloud shows the lowest core mass but manages to accumulate the largest
mass in its margin that dissipates slowly in subsaturated conditions. By comparing the
total cloud mass evolution with the core and margin mass evolutions, it becomes clear
that the total mass is primarily dependent on the cloud core. Another way to see this is
by plotting the core mass fraction (Fig. 3 bottom panel), which shows that clouds are
core dominated (core fraction > 0.5) with respect to mass for most of their lifetimes,
and for all aerosol concentrations.





With respect to cloud total volume, the lower the concentration, the larger the total
cloud volume. We note that the cloud volume here excludes regions of precipitation
below the initial cloud base height. By separating to core and margin regions, one can
see that the total cloud volume is primarily dependent on the volume of the margin,
which increases significantly with decreasing concentration. This is especially true
during the dissipating stages of cloud lifetime, when the cloud is margin dominated.
Although increasing the aerosol concentration does initially yield an increase in core
volume (as was seen for the mass), the extents of the core size are typically smaller
than those of the margin. There are large differences in the relative core volume
percent for the different clouds. The clean (polluted) cloud is margin (core) dominated
with respect to volume for most of its lifetime. Excluding time of formation, the clean
cloud shows the lowest core volume fractions, but manages to maintain its core for the
longest time span.
These results can again be attributed to higher diffusion efficiencies with increase in
aerosol concentration. Additionally, lower collision-coalescence efficiencies also
maintain a narrow droplet spectrum of small droplets in the polluted cloud. During the
growing stage a higher aerosol concentration may permit the cloud to condense more
water, release more latent heat, and promote cloud growth. This explains the larger
core volume sizes. However, after the cloud exhausts its convective potential (i.e. the
growth of the convective core terminates and reaches its peak in mass), its main
method of expansion is by mixing with the environment (i.e. detrainment). We note
that precipitation can also be considered a method of expansion, however our choice
to focus on volume above initial cloud base excludes this effect.  Detrainment results
in sub-saturation conditions and evaporation of LWC. A clear indication for dilution
is seen in Fig. 3 where between 30 and 35 mins of simulation time both the clean and
polluted clouds lose total mass but only the clean cloud increases in total volume. The
polluted cloud evaporates its margin regions efficiently and is thus limited in
detrainment growth. The clean cloud is less efficient in evaporating its margins and
hence can grow by dilution of its LWC upon a larger volume. This large margin
"shields" the core during dissipation stages and enables it the live for a longer time.
The mechanism behind the results in Fig. 3 is demonstrated in Fig. 4, where
horizontal cross-sections of mean (taken in the vertical dimension) cloud RH are
shown for different stages during the clouds' lifetimes. For the polluted cloud, super-



or sub- saturated conditions are rare. The RH throughout the cloud is near 100%
(almost always between 99.8% and 100.2%) except for a few pixels at its far edges
which are a bit below 99%. The polluted cloud resembles what one would expect to
see using a moist adiabatic approximation (i.e. saturation adjustment), where all
excess water vapor above saturation is converted to liquid water, mimicking infinitely
efficient condensation (and evaporation).
The clean cloud shows opposite behavior, with extremes of large super-saturation
during cloud growth (initial stages) and large sub-saturation during cloud dissipation
(final stages). Both extremes can be explained by the low diffusion efficiency in this
case. This enables the clean cloud to expand to larger horizontal extents (by dilution
and mixing with the environment without fully evaporating) and live for longer times.
The intermediate aerosol concentration shows a midway scenario, where the super-
saturation is consumed more efficiently than the clean case and at the same time much
larger values of sub-saturation may exist than those seen for the polluted case.

**4. Results – Cloud field simulations**
In the following section we expand our analyses of aerosol effects on cloud core and
margin from the single cloud scale to the cloud field scale. A cloud field can be
considered as composed of many individual clouds and thus can serve to test the
robustness of the aerosol effects seen for a single cloud. Moreover, cloud fields
include the added complexity of interactions between clouds and the clouds' effects
on their thermodynamic environment.
**4.1.    Sensitivity of different core types to aerosol concentration**
Here CvM space representations (see Sect. 2) are used to observe the core volume
fractions of all clouds in BOMEX cloud field simulations. The rows in Fig. 5
represent different aerosol concentrations while the columns represent different core
type definitions. Different aerosol concentrations produce a vastly different scatter of
clouds in the CvM space, as was previously discussed in depth (Heiblum et al.,
2016a). The clean simulation (25 cm$^{-3}$) shows two disconnected regions of cloud
scatter: one which is adjacent to the adiabatic approximation and one of mainly small
mass and high COG clouds. The former region includes both clouds during their



growth stages (smaller masses, LWP < 10 g m$^{-2}$) and large precipitating entities
(larger masses, LWP > 10 g m$^{-2}$) which form due to merging processes (see (Heiblum
et al., 2016a)). The latter region (small mass and high COG) includes clouds at their
dissipating stage, which form by shedding mechanism off the large cloud entities. We
note also the existence of small mass elements well below the adiabat, representing
precipitation cloud segments which shed off large precipitating clouds.
The polluted simulation (2000 cm$^{-3}$) shows a much more homogeneous scatter of
clouds. The lower part of the scatter (closest to the adiabat) represents the cloud
growing branch while the rest of the scatter represents dissipating clouds, either by
gradual process of rising cloud base or by immediate process of shedding off larger
cloud entity (see Fig. 1, PTI). Precipitating cloud segments below the adiabat are
absent from this simulation. The intermediate simulation (250 cm$^{-3}$) shows a scatter
which generally more resembles the polluted case. However, the existence of
relatively disconnected (from the main cloud scatter) small mass cloud segments
below the adiabat and near the inversion base height resembles the clean simulation as
well.
The results in Fig. 5 show a consistent behavior of the core volume fractions for all
aerosol concentrations, where the $W_{core}$ type shows the largest fractions and the $B_{core}$
type shows the smallest fractions. The $W_{core}$ and $RH_{core}$ generally show a decrease in
core fractions along the growing branch while the $B_{core}$ fraction initially increase with
cloud growth and then decrease for the large mass growing clouds. These results are
consistent with PTI and the single cloud simulations in Sect. 3.1. Nevertheless, some
significant aerosol effects on the partition to core types can be seen. Focusing on the
growing branch first (i.e. clouds located near the adiabat), we note the following:
1) For the $RH_{core}$ type, the core volume fractions of clouds after formation (i.e.

with small mass) increase with decreasing aerosol concentration. This effect

was also seen for the single cloud simulations and can be explained by the

reduced efficiency of super-saturation consumption for fewer aerosols.

2) The $B_{core}$ volume fraction increases at smaller mass values (or earlier in

cloud's lifetime) and to higher values for increasing aerosol concentration.

This effect is complimentary to the previous one, since efficient consumption



of super-saturation should result in more latent heat release and positive
buoyancy.

3) The core volume fractions of the largest mass clouds increase with increasing
aerosol concentration, for all core types.

For the dissipating branch clouds a highly variable pattern of core volume fractions
can be seen, especially for the small mass clouds. For all aerosol concentrations, these
small cloud fragments can be either core dominated, margin dominated, or equally
partitioned. One can assume that these differences can be related to the different
mechanisms by which cloud fragments form, either by gradual dissipation of a large
cloud and by instantaneous shedding of a large cloud. As for aerosol effects on the
dissipating clouds, we see the following:

1) Higher $RH_{core}$ and $W_{core}$ volume fractions for gradually dissipating clouds (by
rising cloud base) with increase in aerosol concentration. This is manifested by
a slower transition from red to blue colors in Fig. 5. It can be explained by the
fact that more aerosols increase the convective intensity and extend the core
size, while efficiently losing the margins, yielding a higher core volume
fraction out of the total cloud.

2) The likelihood to find dissipating cloud fragments with a $B_{core}$ increases with
decrease in aerosol concentration. For the polluted case most of the dissipating
clouds lack a $B_{core}$. This effect was seen in Fig. 1 and explained in Sect. 3.2,
showing that weak downdrafts promote heating and positive buoyancy in low
aerosol concentration cases where evaporation efficiency (and hence cooling)
is limited. This hypothesis is checked for the cloud field scale in Sect. 4.2.

As opposed to the single cloud simulations (Sect. 3) where cloud lifetime can be
easily defined, in cloud field simulations (especially the cleaner cases) many clouds
do not live as individual clouds from formation to dissipation but rather split and
merge with other clouds continuously (Heiblum et al., 2016a). Thus, in order to
evaluate the lifetime evolution of cores in cloud fields, we focus on the growing
branch and use cloud mass [kg] as a proxy for the cloud lifetime during its initial and
mature stages. We assume that in the vicinity of the growing branch a larger mass
corresponds to a later stage in lifetime.



In Fig. 6 the core mass and volume fractions (using the RH definition) of all growing
branch clouds are sorted by mass for the three aerosol concentrations. We note that
the higher cloud masses reached by lower aerosol concentration simulation can be
explained by cloud field organization effects due to precipitation (i.e. increased
merging of clouds) rather than increased cloud condensation (Seigel, 2014;Heiblum et
al., 2016a). The clean case starts off with the highest core fractions (both mass and
volume) which decrease steadily with increase in mass (or increase in lifetime). For
all concentrations, most of the cloud mass is concentrated in the core region. The
polluted case shows a slight increase in mass fractions with increase in mass, while
the other two cases show decreases in mass fractions.
The core volume fractions show lower values than the mass fractions. The clean
clouds are margin dominated for most masses, and the polluted clouds are core
dominated for all masses. The intermediate case is generally confined to values
between the other two cases. Figure 6 can be considered comparable with the lower
panels in Fig. 3, but excluding the dissipating part of those time series. The similar
findings in both figures indicate the robustness of the aerosol effects on core
properties in clouds.
Following the analyses of Sect. 3.1, we next test how aerosol concentration affects the
subset properties of one core type within another for all clouds in a field (Fig. 7). We
focus only on the typically smaller sized cores ($B_{core}$, $RH_{core}$) within larger sized
cores. Out of the three permutations, the $RH_{core}$ inside $W_{core}$ shows the lowest
sensitivity to aerosol. All three growing branches (for the different aerosol
concentrations) consistently show that the $RH_{core}$ is a subset of $W_{core}$ (i.e. $RH_{core} \subseteq$
$W_{core}$) while the dissipation branches show much lower overlap fraction between the
two cores.
Generally, for the dissipating clouds, the lower the mass and the higher the COG, the
smaller the overlap. The dissipating branches do include a scatter of small cloud for
which $RH_{core} \subseteq W_{core}$, comprised of small cloud segments which shed of the main
core regions of larger clouds. These findings slightly differ from those of the single
cloud simulations that show $RH_{core} \subseteq W_{core}$ for their entire lifetimes while for cloud
fields this property breaks downs during dissipation. This difference highlights the
importance of cloud interactions (i.e. splitting, merging) and cloud field air flow





patterns (i.e. organized advection, updrafts, and downdrafts) in determining the
relationships between core types, enabling supersaturation and downdrafts to coincide
in small dissipating clouds.
The other two permutations (i.e. $B_{core}$ inside $RH_{core}$, $W_{core}$) show significant changes
due to aerosol. For the polluted case, $B_{core} \subseteq W_{core}$ for nearly all clouds, including
clouds at initial stages of dissipation. Similar results are seen for $B_{core}$ inside $RH_{core}$,
but with slightly lower pixel fractions. The polluted case thus illustrates the case of
buoyancy production due to convective processes. For the lower aerosol
concentrations, two main aerosol effects are seen:
1) The lower the concentration, the lower the chance that $B_{core}$ is a proper subset

of the other cores for large growing branch clouds.

2) The lower the concentration, the more prevalent the independent dissipating

branch $B_{core}$ that has little to no overlap with the other cores.

For the case of $B_{core}$ within $RH_{core}$, the lower concentrations show an almost binary
scenario where either $B_{core} \subseteq RH_{core}$ or $B_{core} \notin RH_{core}$. These result bear similarity
with the single cloud simulations, where a quick transition (in time) from $B_{core} \subseteq$
$RH_{core}$ to $B_{core} \notin RH_{core}$ was seen. This results implies the existence of two different
buoyancy production processes (as will be shown in Sect. 4.2), one associated with
supersaturation and the other with subsaturation. In contrary, $B_{core}$ inside $W_{core}$,
which shows higher values and more fluctuations in pixels fractions for both single
clouds and clouds fields during dissipation. This is to be expected due to the a more
direct physical link and feedbacks between the $B_{core}$ and $W_{core}$.

**4.2.    Analysis of cloud field buoyancy**
In Sect. 3.2 it was seen that for single clouds, positive buoyancy results from two
main mechanisms: i) convection - where updrafts promote supersaturation and latent
heat release, and thus positive $B_T$ and $B_{Qv}$, and ii) adiabatic heating – where weak
downdrafts promote a positive $B_T$ and neutral $B_{Qv}$. The latter case is dependent on low
evaporation efficiency and hence seen mostly for precipitating stages of low aerosol
concentration simulations.





In Fig. 8 we perform a similar test for the cloud field scale. Instead of analyzing pixel
by pixel, we check whether each buoyancy core within a cloud is $B_T$ or $B_{Qv}$
dominated. To quantify this we use a normalized buoyancy dominance parameter
$\frac{\Sigma pixel_{B_T>0} - \Sigma pixel_{B_{Qv}>0}}{\Sigma pixel_{B>0}}$, where a core comprised of only $B_T>0$ ($B_{Qv}>0$) pixels yields 1
($-1$). Hence, we expect negative (positive) values to indicate dominance of
convective buoyancy (adiabatic heating buoyancy).
Analysis of the buoyancy components in the CvM space (right column, Fig. 8) shows
that the large majority of clouds are $B_{Qv}$ dominated. For all concentrations, clouds
initiate with all pixels showing $B_{Qv}>0$. As clouds develop along the growing branch
the $B_{core}$ becomes more abundant with $B_T>0$ pixels. This is expected with increasing
release of latent heat during cloud growth. During dissipation $B_{Qv}$ again becomes the
dominant component for the majority of clouds. The polluted simulation shows an
extreme case where all buoyancy cores in the simulation are $B_{Qv}$ dominated, while for
the lower concentrations a portion of the dissipating and precipitating clouds are $B_T$
dominated.
Thus, we hypothesize that the polluted simulation only permits buoyancy cores of the
convective type which intersect with the other cores types (i.e. $B_{core} \in RH_{core}, W_{core}$),
while the lower concentrations also permit buoyancy cores of the adiabatic heating
type which do not intersect with the other core types (i.e. $B_{core} \notin RH_{core}, W_{core}$). This
hypothesis is tested by observing the effects of cloud maximum vertical velocity (left
column, Fig. 8) and mean drop size (middle column, Fig. 8) on the relative dominance
of the buoyancy terms. The data is further separated to independent ($B_{core} \notin$
$RH_{core}, W_{core}$) and dependent ($B_{core} \in RH_{core}, W_{core}$) buoyancy subsets of the data.
Clear aerosol effects are seen on cloud mean drop size and maximal W. As expected,
there is a decrease in drop size with increase in aerosol concentration and increase in
maximal velocity. Regarding cloud field buoyancy, as predicted the independent
buoyancy cores are more frequently $B_T$ dominated than the dependent buoyancy
cores.
The polluted case is populated with dependent cores (white scatter) and shows a
classic pre-precipitation convective growth scenario, where relative dominance of the
$B_T$ term increases linearly with increase in cloud mean drop size. A logarithmic
dependence of $B_T$ dominance on maximal W is seen, which saturates at high maximal





W. This can be explained by the fact increased convection mainly increases the
abundance of pixels with $B_T > 0$, but without changing the fact that the entire cloud is
$B_{Qv} > 0$, so that $B_T$ is unlikely to become the dominant term.
The lower concentrations show a more complex scenario. These simulations show a
superposition of dependent core convective growth behavior (i.e. the scatter pattern
seen for the polluted case) and additional populations of both dependent (other white
scatter points) and independent (black scatter) cores. The independent cores span all
the range of possibilities of $B_T$ and $B_{Qv}$ relative dominances. They tend to have larger
cloud mean drop sizes, and near zero maximum W, indicating that they only form at
late non-convective stages of cloud development. The independent cores that are
$B_T$ dominated thus fulfill the characteristics of adiabatic heating process, while the
independent cores that are $B_{Qv}$ dominated may originate from larger clouds (shedding
mechanism) with high humidity content and are slow to evaporate.
The intermediate simulation shows an additional scatter area of dependent core clouds
with increasing of $B_T$ relative dominance for lower maximal W, located between the
independent core clouds and the convective growth core clouds. These clouds may
represent a gradual transition from $B_{Qv}$ dominance to $B_T$ dominance during dissipation
which is only possible in the intermediate simulation. This scatter area is absent from
the clean and polluted simulation. In the former case due to absence of the gradual
dissipation pathway, and in the latter case due to efficient evaporation eliminating
$B_{core}$ during dissipation.

### 4.3.  Aerosol effects on cloud relative humidity

From Fig. 3 it was learned that a large part of the differences in single cloud
characteristics (such as mass, volume, and the partition of these to core and margin
regions) due to aerosols can be attributed to differences in vapor diffusion
efficiencies. In Fig. 9 we check how these aerosol effects are manifested in the cloud
field scale (using the CvM space) by observing the mean relative humidity (RH) in
the cloud core and margin of all clouds. The core mean RH can be taken as a proxy
for condensation efficiency, the margin mean RH as a proxy for evaporation
efficiency. To gain additional intuition regarding the distribution of RH values within



the clouds, vertical cross-sections (parallel to the prevailing wind direction) of the
most massive clouds from each simulation are shown.
The vertical cross-sections demonstrate the large differences in the massive clouds for
each of the simulations. In addition to the increase in precipitation production, lower
aerosol concentrations yield much larger horizontal extents of clouds. The clean,
intermediate, and polluted most massive clouds have a maximum radius of ~ 3, ~ 1,
and ~ 0.5 km, respectively. It is clear from the cross-section that the clean cloud is
actually composed of two large clouds which merge together. For the clean case, the
highest RH values are reached slightly below the cloud top. The edges of the clouds
show sub-saturation conditions, with the lowest RH values observed below the LCL
(precipitation regions) and at the upper interface of the cloud with the environments.
The intermediate case cloud shows lower maximal and minimal RH values and an
increased dominance of the margin region. This cloud penetrates the inversion layer
and entrains dry air into the cloud. In addition, the cloud produces significant
precipitation which initiates downdrafts of dry entrained air through the cloud center.
It can be seen that the increased vertical development of the intermediate case cloud
in comparison with the clean case increases the mixing with the environment. Thus,
the dynamic effect of increased mixing and reduction in cloud RH overcomes the
microphysical effect of increased evaporation and increase in cloud RH. The polluted
case cloud on the other hand shows a homogeneous RH pattern, with most of the
cloud showing around 100% RH and only a thin layer at the cloud edges (mainly at
the upper regions) shows lower RH values. The polluted cloud penetrates the
inversion layer as well, but this case lacks precipitation and the microphysical effect
of evaporation overcomes the dynamical effect of mixing.
Keeping in mind the insights obtained from comparisons of individual cloud, we
move on to compare the RH characteristics of all clouds within the field. Looking first
at core mean RH, a robust decrease is seen with increase in aerosol concentration.
This decrease is seen for all cloud types and locations within the CvM space. The
polluted case displays the most homogeneous pattern with all clouds showing core
mean RH values around 100%, indicating efficient consumption of the
supersaturation. The intermediate case displays a slightly less homogeneous pattern
with values ranging from 100% to 101%, the higher values occurring along the



growing cloud branch, especially for the largest clouds. The clean case shows the
largest variance in core mean RH, ranging from 100% for some cloud fragments that
soon start to dissipate, to 103% in the cores of the large cloud entities. In addition to
the low efficiency in consuming supersaturation, the high RH values in clean large
clouds are due to the "protection" by large margin regions surrounding the core
region.
The CvM patterns of mean margin RH show significant differences between the
polluted case and the other two. The mean margin RH values of the polluted case are
only marginally lower than 100%, since sub-saturated conditions within the cloud are
quickly adjusted by efficient evaporation. Only the largest clouds in the polluted case
permit lower mean margin RH values (~ 95%) due to the entrainment of very dry
environmental pixels near the cloud tops (as seen in the vertical cross-section as well).
The intermediate and clean cases show similar patterns. The smaller mass clouds
(both growing and dissipating) show values above 95%, while the larger mass clouds
show values as low as 85%. The larger clouds are most likely to reach low RH areas
near the inversion base and below the LCL (i.e. sub-cloudy layer) and entrain dry air
and by that reduce the cloud margin RH.
As seen in the vertical cross-section examples, the largest clouds in the intermediate
case have even lower margin RH values than for the clean case. This can be explained
by the increased development of the large intermediate clouds to heights with lower
RH and by more intense downdrafts for these large clouds. The lowest RH values in
the domain are seen for the precipitating fragments (i.e. located below the adiabat).
These fragments typically contain low concentrations of large drop sizes
(precipitation drops) which are slow to evaporate and capable of surviving in low RH
conditions within the sub-cloudy layer.

**Summary**
In this work we explored how the aerosol effects on warm convective clouds are
reflected in their partition to core and margin regions. Following part I of this work
(PTI), we evaluated three types of core definitions: positive buoyancy ($B_{core}$), super-
saturation ($RH_{core}$), and positive vertical velocity ($W_{core}$). Both single cloud and



cloud field models have been used. The former distills the dominant in-cloud
processes affected by aerosols while the latter also takes into consideration the
multiple temporal cloud evolution pathways and the additional effects of cloud field
organization and interactions between clouds.
For all aerosol concentrations, it is shown that the self-contained property of different
core types (i.e. $B_{core} \subseteq RH_{core} \subseteq W_{core}$ is maintained for clouds during their growing
and mature stages. This is especially robust for the $RH_{core} \subseteq W_{core}$ subset. The $W_{core}$
and $RH_{core}$ volume fractions decrease monotonically during cloud growth, while
$B_{core}$ initially increases and then decreases after convection ceases. During growth,
the $RH_{core}$ ($B_{core}$) volume fractions are largest for clean (polluted) clouds. This is due
to low (high) diffusion efficiencies, respectively, where efficient condensation
promotes $B_{core}$ at the expense of the $RH_{core}$.
During dissipation stages cores frequently cease to be subsets of one another and may
either increase or decrease in their volume fractions. In cloud fields we also observe
small cloud fragments which shed off larger cloud entities. This shedding increases
for the lower concentration simulation which produce long-lived large cloud entities.
These fragments show large variance in volume fraction (for all core types)
magnitudes without any consistent behavior. This is due to the fact that they shed off
various locations of the cloud. The polluted, non-precipitating cases, are unique in that
can one expect the $B_{core}$ to decrease monotonically and remain the smallest and a
proper subset of the other cores.
For low aerosol concentration, clouds which are capable of producing precipitation, a
$B_{core}$ may form during dissipation and exist independently of the other core types.
These cores are typically located at the periphery of large clouds, or throughout small
precipitation or dissipating cloud fragments. The increase in $B_{core}$ during dissipation
typically coincides with precipitation production. The fluctuations in $B_{core}$ for low
concentrations may also create a subsequent $W_{core}$, but not of sufficient strength to
also create a $RH_{core}$. Hence, the $RH_{core}$ can be considered the most "well-behaved"
and indicative of cloud lifetime, generally monotonically decreasing in volume
fraction irrespective of aerosol concentration.
We show that the $B_{core}$ in the warm convective cases considered here may form by
two main processes:



1.  Convection: adiabatic cooling within updrafts promotes supersaturation, condensation, and release of latent heat.

2.  Adiabatic heating: weak downdrafts during dissipation or precipitation transport higher potential temperatures from above.

The convective case is seen for all aerosol concentrations, and is characterized by a dependent $B_{core}$ (i.e. $B_{core} \in RH_{core}, W_{core}$). During convection $B_{core}$ pixels have a positive humidity term ($B_{Qv}$), with an increasing abundance of a positive temperature term ($B_T$) pixels with increase in cloud maximum vertical velocities. During dissipation this type of $B_{core}$ shrinks rapidly due to negative $B_T$. The adiabatic heating case is only seen for lower aerosol concentrations, and is characterized by independent $B_{core}$ (i.e. $B_{core} \notin RH_{core}, W_{core}$). In this case $B_T$ is the dominant term in the cloud. The clouds with independent $B_{core}$ experience near neutral vertical velocities for all pixels, and typically show larger cloud mean drop sizes than for the dependent type ones.

The fact that the adiabatic heating $B_{core}$ is absent from polluted clouds highlights the importance of mean drop size and its effect on evaporation rate. The high (low) diffusion (collision coalescence) efficiencies in polluted clouds maintain a small mean drop size and efficient evaporation during entrainment. In PTI we saw that evaporation always has a strong negative effect on buoyancy. In the polluted case the convective $B_{core}$ disappear rapidly during dissipation and cannot form in small cloud fragments even if they experience weak downdrafts. The importance of drop size is illustrated by the fact that even for lower concentrations, clouds with independent $B_{core}$ only exist during late dissipation and precipitating stages after drop size has grown considerably.

Focusing on cores using the RH definition, a cloud's mass (volume) is dependent primarily on the processes in its core (margin). The core increases cloud mass by condensation while the margin increases the cloud's volume by mixing with the environment, or dilution. The magnitude of the effects in each region of the cloud is strongly dependent on the aerosol concentration. Increasing the aerosol concentration increases the vapor diffusion rate, minimizing both the super-saturation and sub-saturation (absolute) values in the cloud. Thus, polluted clouds are efficient in accumulating water mass but also in losing it. This competition between the core mass



gain and margin mass loss regions is what brings about the concept of an optimal
aerosol concentration (Dagan et al., 2015b), and explains why more polluted clouds
are not necessarily more massive.
Polluted clouds are core dominated both in terms of mass and volume, since they can
hardly maintain their margins. Clean clouds are also core dominated in terms of mass,
but to a lesser degree. However, expect for the initial time of cloud formation where
the entire cloud is super-saturated, clean clouds tend to be margin dominated in terms
of volume for most their lifetimes. Thus, despite weaker convection in the clean
clouds, their large, slow evaporating margins enable their cores (and the entire cloud)
to exist for longer time spans by applying a large "protecting shield" around the core.
The different diffusion efficiencies are demonstrated by observing the relative
humidity (RH) values in clouds. Cleaner clouds show larger variance in RH values.
During their growing stages large super-saturation in the core and sub-saturation in
the margin can be seen. During their dissipation stages clouds may exist for minutes
without any cloud core, with the entire cloud at sub-saturation. Polluted clouds show
the opposite, with RH values nearing 100% throughout the cloud, at all stages. Hence,
above a certain aerosol concentration, the saturation adjustment approximation (i.e.
instant condensation of all super-saturation) can be considered valid. However, the
transition from clean to polluted is not always linear. For example, for the largest
clouds in the intermediate case have lower margin RH value than both the clean and
polluted cases. This is due to the fact that the intermediate case manages to develop
taller (than the clean case) clouds with stronger updrafts and downdrafts which entrain
drier air from above the inversion layer base, but at the same time is less efficient in
evaporating (than the polluted case) water and adjusting the RH to 100%.
Finally, we note that the cloud organization also changes with aerosol concentration,
and thus serves as an additional factor affecting the cloud partition to core and margin.
Decreasing the aerosol concentration increases the precipitation yield, which alters the
sub-cloudy layer organization and promotes merging between different clouds (Seifert
and Heus, 2013;Seigel, 2014;Heiblum et al., 2016a). These effects are minimal in the
polluted cases. Hence, to a first approximation polluted cloud fields can be considered
as a superposition of many single clouds while clean cloud fields behave very
differently than a collection of single clean clouds. The continuous merging between



clean clouds creates large cloud entities that evolve along relatively long times. These
large precipitating entities also frequently shed small cloud fragments into the upper
cloudy layer. This effect, combined with the low vapor diffusion, explains why clean
clouds tend to be even more margin dominated (in terms of volume) during growth,
while showing larger core fractions (especially $B_{core}$) during dissipation.
**Acknowledgements**
The research leading to these results was supported by the Ministry of Science &
Technology, Israel (grant no. 3-14444).

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






**Figures**

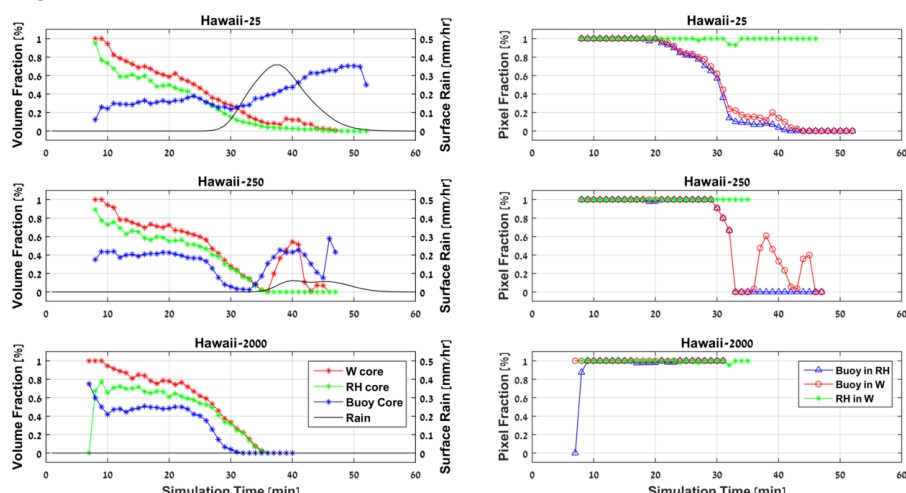


*Figure 1. Left: Time series of core volume fractions ([%], LHS axis) and surface rain-*
*rate ([mm hr$^{-1}$], RHS axis) for the clean (top panel), intermediate (middle panel), and*
*polluted (bottom panel) single cloud simulations. Right: Time series of core pixel*
*fractions within other core types [%], for the respective simulation types. Core*
*volume and pixel fractions are indicated by different line colors (see legends).*






*Figure 2. Scatter plots of pixel total buoyancy [m s⁻²] vs. pixel vertical velocity [m s⁻*
*¹], for the clean (left), intermediate (middle), and polluted (right) simulations. Data*
*includes all cloudy pixels during all time steps. Colors represent magnitude of*
*buoyancy temperature term ($B_T$, upper row) and humidity term ($B_{Qv}$, lower row),*
*where red (blue) shades indicate positive (negative) values. Markers with black dots*
*superimposed represent temporal stages with non-zero surface precipitation. White*
*arrows indicate outlier scatter of pixels with positive buoyancy and negative vertical*
*velocity.*

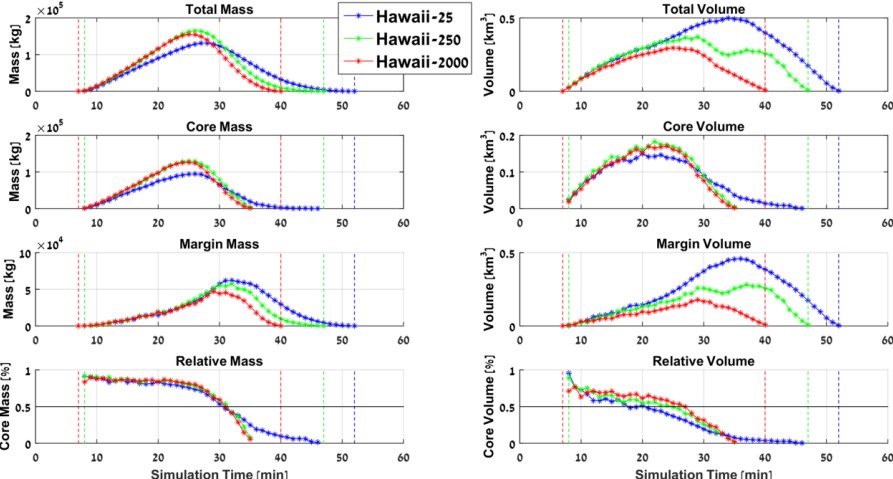


*Figure 3. Time series of cloud mass ([kg], left column) and cloud volume ([km³], right*
*column) for the different aerosol concentrations simulations (see legend). The total,*
*core, margin, and relative fraction values are shown for each parameter, as indicated*
*by panel titles. The core here is defined according to RH>100% definition.*





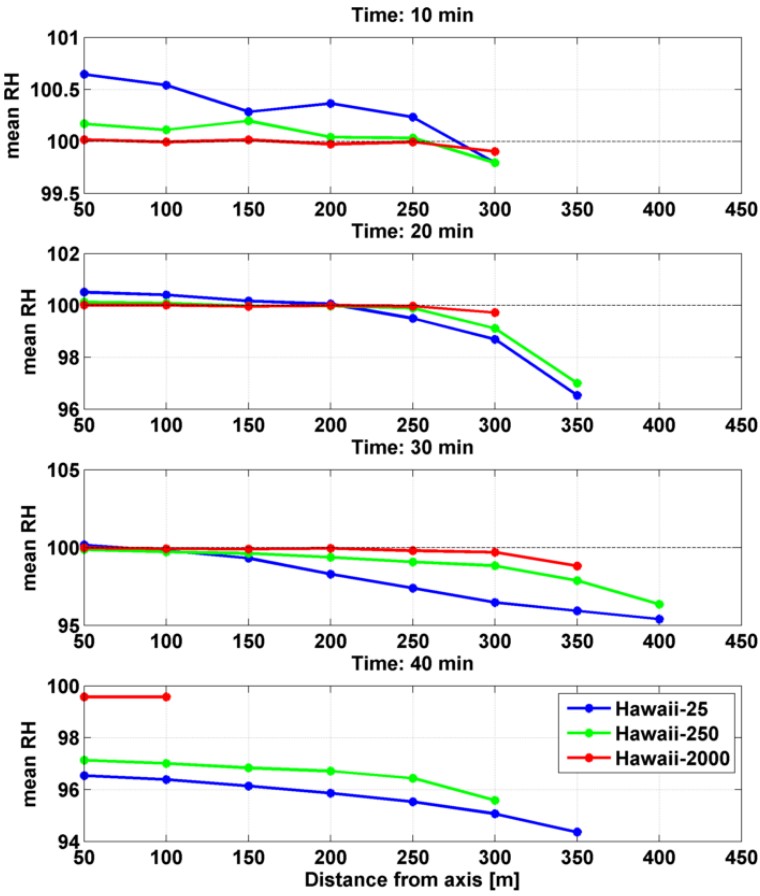


*Figure 4. Four temporal snapshots (see panel titles for times) of RH [%] horizontal*
*cross-sections. Panels include the results of different aerosol concentrations (see*
*legend). Cross-sections are obtained by taking the mean RH of all vertical levels for*
*each horizontal distance from the cloud center axis.*






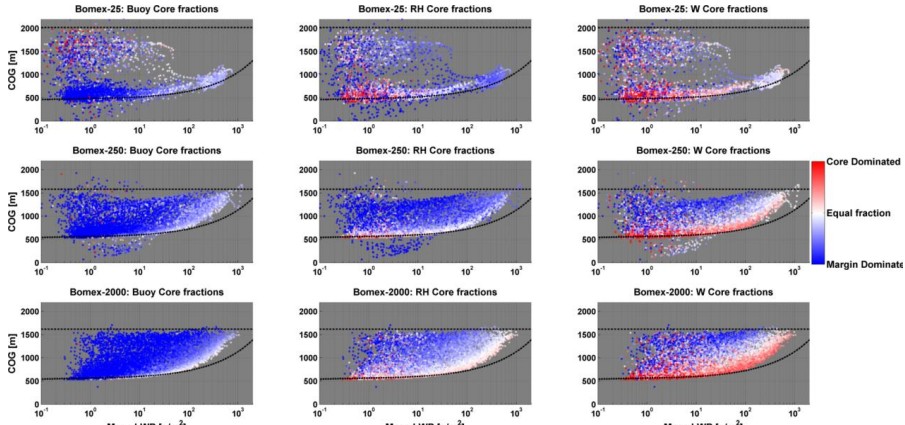

*Figure 5. CvM phase space diagrams of $B_{core}$ (left column), $RH_{core}$ (middle column), and $W_{core}$ (right column) volume fractions for all clouds between 3 h and 8 h in the BOMEX simulations. The upper, middle, and lower panels correspond to the clean, intermediate, and polluted aerosol cases. The red (blue) colors indicate a core volume fraction above (below) 0.5. The majority of clouds are confined to the region between the adiabatic cloud growth approximation (curved dashed line) and the inversion layer base height (horizontal dashed line). For an in-depth description of CvM space characteristics, the reader is referred to Sect. 2.4 in PTI.*

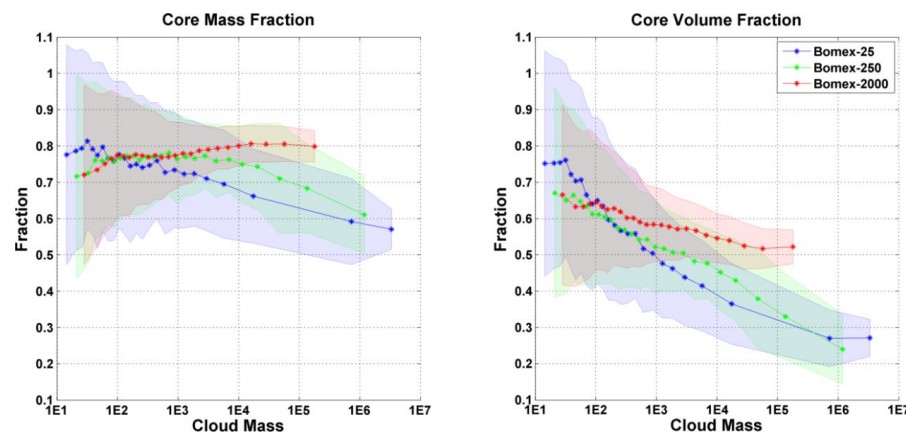

*Figure 6. Average core mass fraction (left) and volume fraction (right) values for different aerosol concentrations, as indicated in the legend. The average only includes*



*growing branch clouds from within the CvM space (i.e. clouds located in proximity to*
*the adiabat). The core here is defined according to RH>100% definition.*


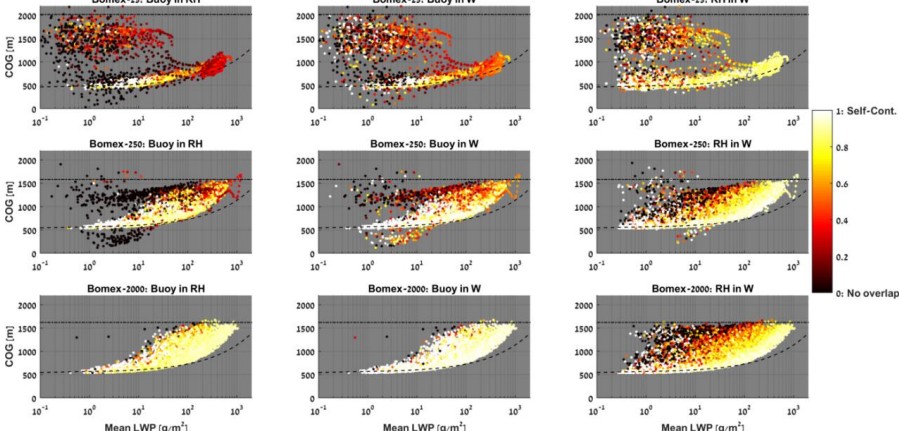


*Figure 7. CvM space diagrams showing the pixel fractions of $B_{core}$ within $RH_{core}$,*
*$W_{core}$, and $RH_{core}$ within $W_{core}$ (as indicated in the panel titles). Bright colors*
*indicate high pixel fractions (large overlap between two core types) while dark colors*
*indicate low pixel fraction (little overlap between two core types). The differences in*
*the scatter density and location for different panels are due to the fact that only clouds*
*which contain a core fraction above zero (for the core in question) are considered.*


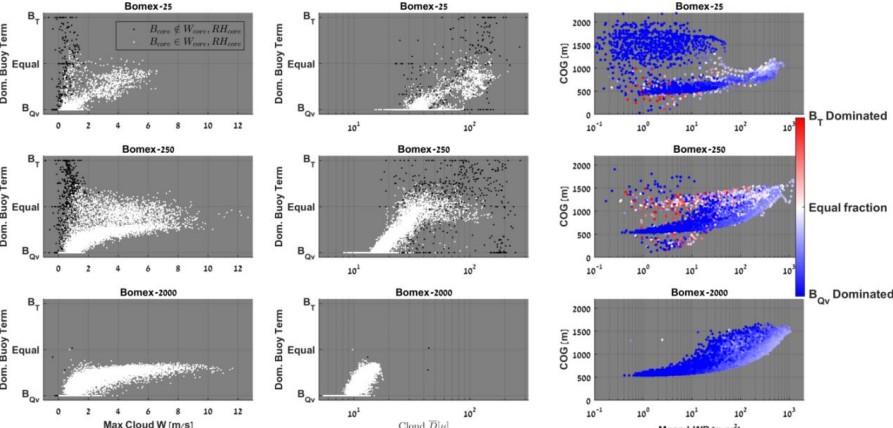






*Figure 8. Analysis of dominant buoyancy term within $B_{core}$ of clouds (see text for*
*details). As seen in previous figures, rows represent clean (top), intermediate*
*(middle), and polluted (bottom) simulations. Left: dependence on maximum vertical*
*velocity within cloud. Middle: dependence on partition of total cloud mass to cloud*
*droplets and rain drops. Right: CvM space diagrams of all clouds with $B_{core}$, where*
*red (blue) shades indicate temperature (humidity) buoyancy terms dominate the*
*cloud.*


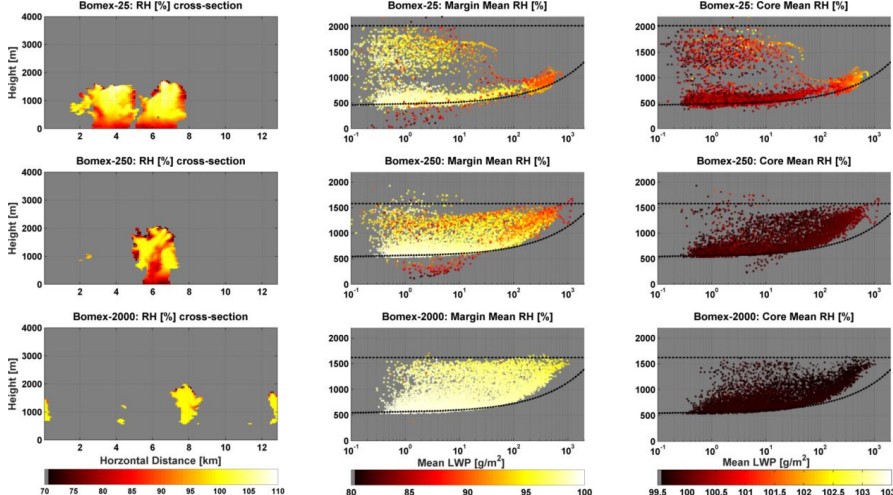


*Figure 9. Left column – Relative Humidity (RH [%]) vertical cross-sections slicing*
*through the center of gravity of the most massive cloud in each simulation. Middle*
*and right columns display CvM space diagrams of mean cloud margin RH and mean*
*cloud core RH, respectively, using the $RH_{core}$ definition. The upper, middle, and*
*lower panels correspond to the clean, intermediate, and polluted aerosol cases (see*
*panel titles). Notice the different color bar ranges for margin and core mean RH*
*panels.*
