# Peer review of "Manuscript under review for journal Atmos. Chem. Phys."

_Atmospheric Chemistry and Physics, 2018_

## Referee Comment (RC1) · Anonymous Referee #1 · 26 Oct 2018

Review of: "Core and margin in warm convective clouds: Part II: aerosol effects on core properties."

Authors: Reuven Heiblum, Lital Pinto, Orit Altaratz, Guy Dagan, Ilan Koren

General comment:

This paper follows from Part I which sought to examine the various methods of defining the cores and margins of convective clouds by using buoyancy, RH, and vertical velocity to define the core. They showed that these core diagnostics can be subsets of one another, but that this varies in space and time. This follow-on study examines the impacts of varying the aerosol concentration on the core definitions. Given that aerosols can change the cloud DSD, the condensation/evaporation rates can change,

and thus the field RH, latent heating, and W. The authors effectively demonstrate the aerosol effects on the evolution of the convective cores and margins.

Two main concerns that can readily be addressed are the need to: (1) better state the goals and hypotheses of the study and state what makes this study novel compared to similar ones in the literature, and (2) better reference past studies in the introduction relative to many of the scientific statements that are made regarding aerosol effects on cloud droplets.

Specific comments:

1. Line 72-73: The warm phase convective invigoration process has brought about some lively debate in the community in recent years. It does seem however that lately more papers are being published on the matter. I would suggest adding a few additional references that may include the following:

Sheffield, A.M., S.M. Saleeby, and S.C. van den Heever, 2015: Aerosol-induced mechanisms for cumulus congestus growth. J. Geo. Res., 120, 8941-8952.

Saleeby, S.M., S.R. Herbener, S.C. van den Heever, and T.S. L'Ecuyer, 2015: Impacts of cloud droplet-nucleating aerosols on shallow tropical convection. J. Atmos. Sci., 72, 1369-1385.

2. Lines 80-82: Perhaps add more recent references regarding impacts of aerosol and smaller cloud droplets on condensation and evaporation rates in clouds and along cloud edges.

Grant, L.D., and S.C. van den Heever, 2015: Cold pool and precipitation responses to aerosol loading: modulation by dry layers. J. Atmos. Sci., 72, 1398-1408.

Storer, R.L., and S.C. van den Heever, 2013: Microphysical processes evident in aerosol forcing of tropical deep convective clouds. J. Atmos. Sci.,70,430-446.

Saleeby, S.M., S.R. Herbener, S.C. van den Heever, and T.S. L'Ecuyer, 2015: Impacts

of cloud droplet-nucleating aerosols on shallow tropical convection. J. Atmos. Sci., 72, 1369-1385.

3. Lines 86-88: This would be a good place to address the concept of "microphysical buffering". This has become much more prominent of a concept in the past few years.

4. Lines 97-99: Figure 17 of Khain et al. (2008, JAS) addresses this aerosol impact on various cloud types and cloud systems. This should be referenced.

5. Lines 100-110: This paragraph should cite Reutter et al. (2009) with respect to the "aerosol-limited" vs "updraft-limited" regimes. Looks like you already have this paper in your reference list, but it would be good to add the citation in this paragraph.

6. Line 140: Here you are transitioning from the paper introduction to the methods section. Your introduction is very thorough, but you haven't yet stated the goals or science questions of the paper. Please make sure you tell your audience the purpose of the paper and why it is important and novel.

7. Lines 239-244: Things get a bit confusing when you refer to precipitation and evaporation of droplets. It's not clear if you're referring to "cloud droplets" or "rain drops". It would be helpful if cloud hydrometeors are always referred to as "droplets" and rain (precipitation) as "drops". So, are you indicating here that the clean case leads to larger cloud droplets and larger rain drops?

8. Lines 308-316: This is just a comment, but I appreciate your analysis here and how you allude to polluted clouds essentially mimicking a saturation adjustment with respect to condensation, and how clean clouds allow substantial supersaturation to be carried about. Given that saturation adjustment schemes are still often used in microphysics parameterizations, this re-emphasizes that use of such a scheme can be very inappropriate except under specific circumstances.

9. Section 4: Moving into this section reminded me to ask about how your aerosols are treated in the model following initialization. Are the initial aerosol concentrations

homogeneous in 3D, do the aerosols advect around the domain, are aerosols removed upon nucleation and regenerated upon droplet evaporation? Are there aerosol sources and sinks? This could certainly be of most importance in a field of clouds.

10. Figure 5: Why is the inversion layer base height higher in the clean case? I don't recall this being addressed in the paper. Is it initially the same in all cases and then changes over time due to microphysical and dynamical interactions?

11. Line 579: When you refer to evaporation throughout the paper are you referring to partial evaporation as a process or fully evaporated droplets? This could be clarified a bit in the paper. Many past papers including some cited herein often refer to net evaporation of drops/droplets as a distribution without specific concern for full evaporation of droplets.

12. Summary section: I find the summary to be a bit over-comprehensive. It's helpful to the reader to keep this concise and to the point. Keep to the main conclusions and main mechanisms. Details can be seen in the bulk of the paper.

Technical corrections:

Figures: My main comment about the figures is that most of the them need to be larger, especially the fonts, so that they are easily readable.

Line 302: Should read as "and enables it to live for..." Line 426: Should read as "segments which shed off the main..." Line 449-451: I find the wording here to make the sentence confusing. Please try clarifying this sentence. Line 521-522: This is a bit of a run-on sentence with a comma separator. Line 656: This should read as "However, except for the..."

---

## Referee Comment (RC2) · Anonymous Referee #2 · 27 Mar 2019

In the study effects of aerosols on the structure of small CU are investigated by analyzing the results of axisymmetric cloud model (single cloud simulations), as well as model of cloud ensemble (SAM) (for investigation of general properties of cloud ensembles). To shorten number of parameters, cloud averaged properties are analyzed. Clouds are characterized by core and margin, and effects of aerosols on these regions are investigated. The paper is of interest.

However, major revision is necessary.

The comments and remarks are presented below.

1. Figures 5, 7, 9 indicate that simulation with low aerosol concentration was performed for inversion base at 2000 m, while in other simulations the inversion base was

at 1500m. Since the cloud dynamical and microphysical structure as well as cloud size depend on the inversion height, the comparison of the aerosol effects should be performed under similar thermodynamic background conditions.

2. The terminology used in the study is not widely accepted and needs better definition. For instance, it is necessary mathematically define what is "condensation efficiency", "diffusion efficiency", etc. (It is possible that such definitions are in Pt 1 of the study. Nevertheless, they should be defined in the present study as well). Note that in addition to equation of diffusion growth ("diffusion efficiency"), there is a turbulent diffusion.

3. Different definitions of cloud cores and cloud margin are interesting. At the same time, these definitions do not agree with the accepted ones. Such definitions lead to a paradox that small dissipating clouds may contain cloud cores. Supposedly, some minimum LWC value should be included into the definition. This will exclude cases, when dissolving cloud with negligible LWC is still considered as combination of cloud core and cloud margin. Below more detailed comments are presented. line 24 Abstract. The values Bcore, Rcore, W core are not defined yet and should be either defined or excluded from the abstract. Line 45. The text reads: "detrainment while losing mass". In cloud physics detrainment is "large scale" outflow, typically near cloud top. Interaction of clouds with environment is characterized by entrainment and mixing. The authors supposedly mean that small cloud volumes leaving the parent cloud loss their mass by evaporation.

Line 57. what difference of DSD do you mean? Line 58. The sentence is not clear or not correct. The nucleation itself that takes place at rN>rNcrit does not accompanied by decrease of S. S decreases as a result of diffusional growth of nucleated droplets.

Line 77. Strictly speaking, the diffusion growth equation is not symmetric with respect of processes of condensation/evaporation. This asymmetry is considered, sometimes, as a mechanism of DSD broadening (e.g. Korolev, 1995, JAS, 52, 3620-3634).

Line 161. Comment concerning the cloud core definitions. The condition W>0 takes

place in cloud cores of devolving clouds. Since the time period of cloud developing is relatively short, effects of mixing with surrounding air may not be significant (depending of cloud size and W). At dissolving stage, W<0. So, there is no cloud core in your definition. At the same time LWC in the cloud may has obvious maximum in the cloud center (interior)

line 184. How can be explained updrafts at B<0? Gravity waves?

Line 200. In such case it is difficult to call the condition W>0 as definition of cloud core. Small positive W can take place all over cloud just by turbulent fluctuations. Katzwinkel et al. (2014) and Schmeissner et al. 2015 determine cloud interior as LWC>0.2 gm-3 this condition guaranties that the region chosen is in the cloud interior.

line 202. Fig 1. Figure caption. Define LHS axis, RHS axis. What is "other core types"?

lines 236-240 The mechanism proposed requires additional justification. The another option is that in low CCN case drizzle and rain drops rapidly fall down, so LWC is very low in the subsiding of the air. Another possible mechanism is turbulent mixing between warm core and colder margin air. This mixing should lead to an increase in T, i.e to appearance of positive buoyancy. Can you justify the mechanism that is proposed in the study?

What is the cloud stage? Developing or dissolving? Do you see this effect at cloud center or cloud periphery?

Line 248 How is cloud margin region defined and calculated?

line 254. It is difficult to see the non-monotonic dependence. We see that the maximum cloud mass takes place at high CCN concentration, but lifetime is larger for the low CCN concentration case.

line 285. I do not fully agree with the interpretation. In case of high CCN concentration droplets are small and mixing with surrounding leads to fast complete evaporation of the droplets. Moreover, small droplets fully and easily evaporate also at W<0. At the

same time, larger droplets formed at low CCN concentration evaporate only partially. Why is it necessary to focus on the weak effect of the differences in the evaporation rates?

line 293 At the dissolving stage cloud air descends, i.e. W<0 within cloud body (Schmeisner et al. 2015). The subsiding dramatically decreases RH and leads to droplet evaporation. It is natural, that small droplets evaporate first. This decreases the life time of clouds in polluted air. It would be important to separate two effects: turbulent mixing of clouds with surrounding and their evaporation at W<0. Note that small Cu often dissipate and evaporate within the inversion layer, where turbulence (i.e. mixing is weak). In such case, namely subsiding place dominating role in cloud dissolving.

line 295. Which effect? How can precipitation be considered as a method of...? please reword the sentence.

What kind of expansion can be induced by precipitation? Why the "choice to focus on volume above initial cloud base excludes this effect"? If the precipitation-induced cooling leads to the formation of new clouds, it is impossible to exclude the effect by the choice of the altitude, above which cloud properties are considered.

line 295 Detrainment is the outflow of cloud mass from the cloud. It cannot change the cloud properties. Entrainment of dry environment air into the cloud indeed can lead to subsaturation. Regular (non-turbulent) entrainment takes place near cloud base. At later cloud edges lateral turbulent entrainment and mixing takes place.

line 298. The effect of dilution depends on cloud width. The larger width the lower the effect of lateral mixing is. The increase of decrease of cloud width depends also on LWC. line 300. What is "detrainment growth"?

line 314. To define "diffusion efficiency".

line 327. It is interesting to see the RH (r) profiles in the humid shell around cloud.

line 375. Do you suppose that dissipating clouds may contain dominating cores? How does it agree with observations?

Line 375. Figure 5 shows that simulations with low CCN concentration were performed for the case of 2000 m inversion altitude. Two other simulations were performed for 1500 m altitude. The clouds should be quite different geometrically and microphysically in such cases. How can such clouds be compared?

Line 417. 1) we see again that there the difference in the inversion level in the simulations. Higher clouds can have larger cloud cover and longer life time etc. 2) It is necessary to add to the figure caption the conditions corresponding to the rows and columns or refer notations in fig 5.

Line 458. I still wonder, how weak downdraft can lead to the temperature of subsiding air higher than in surrounding. Such subsidence should be actually along the moist adiabate. Why the downdrafts should be week? It seems that subsidence accompanied by evaporation leads to cold pool that accelerates formation of new clouds. It seems that positive buoyancy in the area of weak downdraft is the results of horizontal mixing between warm zone with W>0 (with high buoyancy) and the cloud periphery.

line 531. One can suppose that many clouds are isolated even in the clean case. Why do you illustrate the clean case by merging clouds?

Line 572. So, the formation of low RH is the result of averaging over wider layers which contain the inversion layer and layer below LCL. Please confirm.

Line 619. Term convection is not suitable. Besides, if you want to compare T with surrounding, moist adiabatic cooling results in heating as compared with the surrounding.

Line 621. Adiabatic heating is also not exact term. In the situation considered adiabatic heating is accompanied by turbulent mixing and droplet evaporation. So, many factors determine T in this area, so process is not adiabatic.

---

## Author Comment (AC1) · 7 Jun 2019

**Reply to reviewers' comments on** *"Core and margin in warm convective clouds.*

*Part II: aerosol effects on core properties"*

## Reply to reviewer #1

**General Comment**

This paper follows from Part I which sought to examine the various methods of defining the cores and margins of convective clouds by using buoyancy, RH, and vertical velocity to define the core. They showed that these core diagnostics can be subsets of one another, but that this varies in space and time. This follow-on study examines the impacts of varying the aerosol concentration on the core definitions. Given that aerosols can change the cloud DSD, the condensation/evaporation rates can change, and thus the field RH, latent heating, and W. The authors effectively demonstrate the aerosol effects on the evolution of the convective cores and margins.

**Main Comments:**

Two main concerns that can readily be addressed are the need to:

MC1) Better state the goals and hypotheses of the study and state what makes this study novel compared to similar ones in the literature.

MA1) A paragraph was added to the introduction describing the novelty and goals of the paper: "***As a continuation to Part I of this work (hereafter PTI), in this part we***

***analyze aerosols effects on the cloud's partition to core and margin throughout the***

***lifetime of a cloud. We report the consequences these effects have on evolution of a***

***cloud, in terms of volume, mass, and lifetime. As opposed to other works that typically***

***focus on a single cloud core definition, here three different definitions are used (see***

***Sect. 2), with emphasis put on the sensitivity of each core definition to aerosol***

***concentration. Moreover, the combination of single cloud with large eddy***

***simulations enables us to gain process level understanding and test the robustness of***

***our findings.***".

MC2) Better reference past studies in the introduction relative to many of the scientific statements that are made regarding aerosol effects on cloud droplets.

MA2) We thank the reviewer for all the suggestions of additional references to this work, which were all implemented into the revised manuscript as described in the specific comments below.

**Specific Comments:**

SC1) Line 72-73: The warm phase convective invigoration process has brought about some lively debate in the community in recent years. It does seem however that lately more papers are being published on the matter. I would suggest adding a few additional references that may include the following:

Sheffield, A.M., S.M. Saleeby, and S.C. van den Heever, 2015: Aerosol-induced mechanisms for cumulus congestus growth. J. Geo. Res., 120, 8941-8952.

Saleeby, S.M., S.R. Herbener, S.C. van den Heever, and T.S. L'Ecuyer, 2015: Impacts of cloud droplet-nucleating aerosols on shallow tropical convection. J. Atmos. Sci., 72,

1369-1385.

SA1) We thank the reviewer for these additional references. They are now added to the text: "***The processes described above enable the more polluted cloud to condense***

***more water and intensify its growth via increased release of latent heat (Kogan and***

***Martin, 1994; Koren et al., 2014; Saleeby et al., 2015; Sheffield et al., 2015).".***

SC2) Lines 80-82: Perhaps add more recent references regarding impacts of aerosol and smaller cloud droplets on condensation and evaporation rates in clouds and along cloud edges.

Grant, L.D., and S.C. van den Heever, 2015: Cold pool and precipitation responses to aerosol loading: modulation by dry layers. J. Atmos. Sci., 72, 1398-1408.

Storer, R.L., and S.C. van den Heever, 2013: Microphysical processes evident in aerosol forcing of tropical deep convective clouds. J. Atmos. Sci.,70,430-446.

Saleeby, S.M., S.R. Herbener, S.C. van den Heever, and T.S. L'Ecuyer, 2015: Impacts of cloud droplet-nucleating aerosols on shallow tropical convection. J. Atmos. Sci., 72,

1369-1385.

SA2) Again, we thank the reviewer for bringing these references to our attention. We added them to the revised text: "***An opposite effect should take place in the sub***

***saturated regions of the cloud, where more numerous and smaller droplets increase***

*the evaporation rate and loss of cloud mass (Grant and van den Heever, 2015;*
*Saleeby et al., 2015; Storer and van den Heever, 2013).*"

SC3) Lines 86-88: This would be a good place to address the concept of "microphysical
buffering". This has become much more prominent of a concept in the past few years.
SA3) The concept of microphysical buffering was added to the text, as follows: "*A*
*different approach to aerosol effects suggests that cloud systems are buffered to*
*microphysical effects (Stevens and Feingold, 2009). Several studies have shown that*
*given enough time for the cloud system to reach steady state, cloud macro-physical*
*parameters (e.g. cloud fraction, rain yield) show similar results for various aerosol*
*concentrations (Carrió and Cotton, 2014; Glassmeier and Lohmann, 2018; Seifert et*
*al., 2015)."*

SC4) Lines 97-99: Figure 17 of Khain et al. (2008, JAS) addresses this aerosol impact
on
various cloud types and cloud systems. This should be referenced.

SA4) Thank you for this suggestion, the reference was added to the relevant line.

SC5) Lines 100-110: This paragraph should cite Reutter et al. (2009) with respect to
the "aerosol-limited" vs "updraft-limited" regimes. Looks like you already have this
paper in your reference list, but it would be good to add the citation in this paragraph.

SA5) Agreed, the reference was added to the relevant paragraph as follows: "*Based on*
*the idea that clouds can be partitioned to aerosol-limited, updraft-limited, or aerosol*
*and updraft sensitive regimes (Reutter et al., 2009), a unified theory for the*
*contradicting results regarding aerosol effects was suggested (Dagan et al., 2015)".*

SC6) Line 140: Here you are transitioning from the paper introduction to the methods
section. Your introduction is very thorough, but you haven't yet stated the goals or
science questions of the paper. Please make sure you tell your audience the purpose
of the paper and why it is important and novel.

SA6) Thank you for this comment, we have added a paragraph explaining the goals of this paper (see MA1 above).

SC7) Lines 239-244: Things get a bit confusing when you refer to precipitation and evaporation of droplets. It's not clear if you're referring to "cloud droplets" or "rain drops". It would be helpful if cloud hydrometeors are always referred to as "droplets"

and rain (precipitation) as "drops". So, are you indicating here that the clean case leads to larger cloud droplets and larger rain drops?

SA7) In these sentences we are referring to the entire distribution of drops, which is skewed to larger sizes in clean cases. For clarity, we switched the word droplet to the more general word drop: "***Moreover, the occurrence during precipitating stages and***

***for lower aerosol concentrations indicates that slow evaporation due to larger drop***

***sizes is crucial.***".

SC8) Lines 308-316: This is just a comment, but I appreciate your analysis here and how you allude to polluted clouds essentially mimicking a saturation adjustment with respect to condensation, and how clean clouds allow substantial supersaturation to be carried about. Given that saturation adjustment schemes are still often used in microphysics parameterizations, this re-emphasizes that use of such a scheme can be very inappropriate except under specific circumstances.

SA8) Thank you for the comment. We have seen such mimicking effects in previous work as well (Heiblum et al., 2016b).

SC9) Section 4: Moving into this section reminded me to ask about how your aerosols are treated in the model following initialization. Are the initial aerosol concentrations homogeneous in 3D, do the aerosols advect around the domain, are aerosols removed upon nucleation and regenerated upon droplet evaporation? Are there aerosol sources and sinks? This could certainly be of most importance in a field of clouds.

SA9) Thank you for this important comment. To answer your questions, aerosols are initialized homogeneously in 2D (horizontally), maintaining constant mixing ratio with height. They are advected around the domain, and are removed upon nucleation and regenerated upon evaporation. Wet scavenging serves as a sink, while there are no sources. We added some details about how the model treats aerosols and a relevant reference (Heiblum et al., 2016a) for a more complete description: "*To study the effects of aerosols on the cloud cores we run each model setup with three different aerosol concentrations: clean – 25 cm$^{-3}$, intermediate – 250 cm$^{-3}$, and polluted – 2000 cm$^{-3}$. The model domain is initialized using an oceanic size distribution (Altaratz et al., 2008; Jaenicke, 1988), maintaining constant mixing ratio with height. Aerosol budget includes removal by nucleation and regeneration upon evaporation, while wet scavenging by precipitation removes aerosols from the domain. Thus, the aerosol concentration may be depleted by 20%–40% (depending on the precipitation amount) during the simulation. More on the treatment of aerosols in the cloud field model can be found in previous work (Heiblum et al., 2016a).*".

SC10) Figure 5: Why is the inversion layer base height higher in the clean case? I don't recall this being addressed in the paper. Is it initially the same in all cases and then changes over time due to microphysical and dynamical interactions?

SA10) As guessed correctly by the reviewer, the inversion base height is initially the same for all cases and evolves differently with time. We now address this point in the paper: "*It should be noted that horizontal dashed lines in Fig. 5 represent the inversion base height after 5 hours of simulation (approximately middle of simulation), where an increase in the inversion base height is seen with decrease in aerosol concentration. This is due to increased net warming in the upper cloudy layer (i.e., release of latent heat during condensation with reduced local evaporation) with increase in precipitation (Dagan et al., 2016; Heiblum et al., 2016b) , which raises the inversion base*".

SC11) Line 579: When you refer to evaporation throughout the paper are you referring to partial evaporation as a process or fully evaporated droplets? This could be clarified a bit in the paper. Many past papers including some cited herein often refer to net evaporation of drops/droplets as a distribution without specific concern for full evaporation of droplets.

SA11) Thank you for this comment that helped us clarify this point. Throughout the paper we refer to evaporation as a process (i.e. mass evaporated per second [g/s]), and hence many times mention evaporation rates rather than how many droplets were fully evaporated. We added a short sentence to the introduction to clarify this point:

*"Henceforth evaporation will be referred to as a process (i.e. change of mass per unit*

*time) rather than complete evaporation of a water drop."*

SC12) Summary section: I find the summary to be a bit over-comprehensive. It's helpful to the reader to keep this concise and to the point. Keep to the main conclusions and main mechanisms. Details can be seen in the bulk of the paper.

SA12) After rereading the summary, we agree with this comment. The summary has been shortened considerably.

SC13) Figures: My main comment about the figures is that most of the them need to be larger, especially the fonts, so that they are easily readable.

SA13) All the figures were redone so that the fonts are larger and readable.

SC14) Line 302: Should read as "and enables it to live for…"

SA14) Thank you, the suggested correction was carried out.

SC15) Line 426: Should read as "segments which shed off the main…"

SA15) Thank you, the suggested correction was carried out.

SC16) Line 449-451: I find the wording here to make the sentence confusing. Please try clarifying this sentence.

SA16) The sentence was rephrased as follows: "*In contrary, pixels fractions of Bcore*

*inside Wcore span the entire range of values (i.e. partial overlaps between the core*

*types), as seen for both single clouds and cloud fields during dissipation*"

SC17) Line 521-522: This is a bit of a run-on sentence with a comma separator.

SA17) We have changed the sentence as follow: "*In Fig. 9 we check how these aerosol*

*effects are manifested in the cloud field scale (using the CvM space) by observing the*

*mean relative humidity (RH) in the cloud core and margin of all clouds, where the*

*core (margin) mean RH can be taken as a proxy for condensation (evaporation)*

*efficiency*".

SC18) Line 656: This should read as "However, except for the…

SA18) Thank you, we have reformulated the sentence: "*However, excluding the initial*

*time of cloud formation where the entire cloud is super-saturated, clean clouds tend*

*to be margin dominated in terms of volume for most their lifetimes*".

**Reply to reviewer #2**

**General Comment**

In the study effects of aerosols on the structure of small CU are investigated by analyzing the results of axisymmetric cloud model (single cloud simulations), as well as model of cloud ensemble (SAM) (for investigation of general properties of cloud ensembles). To shorten number of parameters, cloud averaged properties are analyzed. Clouds are characterized by core and margin, and effects of aerosols on these regions are investigated. The paper is of interest. However, major revision is necessary.

**Main Comments:**

MC1) Figures 5, 7, 9 indicate that simulation with low aerosol concentration was performed for inversion base at 2000 m, while in other simulations the inversion base was at 1500m. Since the cloud dynamical and microphysical structure as well as cloud size depend on the inversion height, the comparison of the aerosol effects should be performed under similar thermodynamic background conditions.

MA1) Thank you for this important comment that helped us clarify our method. All simulations were initialized using the same thermodynamic profile. Figs. 5, 7, 9 show the inversion base height after 5 hours of simulation (and not the initial state), and thus are not equal for different simulations because of microphysical and dynamical interactions between the clouds and their environment that modify the temperature profile of the domain. We choose to display this inversion height (rather than the initial one which is equal for all simulations) since it better reflects the CvM space cloud scatter of the entire simulation. We added to the revised version an explanation clarifying these differences: "*It should be noted that horizontal dashed lines in Fig. 5*

*represent the inversion base height after 5 hours of simulation (approximately middle*

*of simulation), where an increase in the inversion base height is seen with decrease*

*in aerosol concentration. This is due to increased net warming in the upper cloudy*

*layer (i.e., release of latent heat during condensation with reduced local evaporation)*

*with increase in precipitation (Heiblum et al., 2016b), which raises the inversion*

*base".*

MC2) The terminology used in the study is not widely accepted and needs better definition. For instance, it is necessary mathematically define what is "condensation efficiency", "diffusion efficiency", etc. (It is possible that such definitions are in Pt 1

of the study. Nevertheless, they should be defined in the present study as well). Note that in addition to equation of diffusion growth ("diffusion efficiency"), there is a turbulent diffusion.

MA2) We have reviewed the terminology used in the work and defined it when necessary. For the example of diffusion efficiency: "*We note that throughout this work*

*the word* **efficient** *will be used to describe both the rate and the total change of mass*

*attributed to a microphysical process.*". This definition is based on the multiple references and descriptions listed in the introduction section.

MC3) Different definitions of cloud cores and cloud margin are interesting. At the same time, these definitions do not agree with the accepted ones. Such definitions lead to a paradox that small dissipating clouds may contain cloud cores. Supposedly, some minimum LWC value should be included into the definition. This will exclude cases, when dissolving cloud with negligible LWC is still considered as combination of cloud core and cloud margin.

MA3) As reported in Part I, we do take a minimum threshold of 0.01 g/kg for definition of a cloudy pixel. As a matter of fact, we were questioned about the LWC threshold during the review process of Part I, and showed that this definition best captures the main cloudy processes of condensation and evaporation. We quote our answer here.

"*The question of cloud pixel liquid water content (LWC) threshold is something we*

*have examined as part of this work. We started by taking an even lower threshold of*

*0.005 g/kg (Cohen and Craig, 2006) but eventually raised the threshold to 0.01 g/kg*

*based on other works (Jiang et al., 2009; Xue and Feingold, 2006). The impact of*

*threshold choice is shown in Fig. RA1 below. The 0.01 and 0.005 g/kg thresholds*

*yield similar results with regards to cloud volume, while higher thresholds (0.05 and*

*0.1 g/kg) reduce cloud volume significantly. By taking areas of condensation and*

*evaporation as indicators of cloudy regions, it can be seen that the higher values*

*thresholds "miss" pixels with high evaporation rate (vapor diffusion), in both*

*growing and dissipating stages of cloud lifetime. Hence, we find that the 0.01 g/kg*

*threshold best reflects a cloudy volume, without the risk of including insignificant*

*cloud debris as can be seen in some cases for the lower 0.005 g/kg threshold*.".

[Figure]

*Fig. RA1. Four vertical cross-sections (at t=8, 20, 30, 40 minutes) during the single*

*cloud simulation with aerosol concentration of 500 CCN. Y-axis represents height [m]*

*and X-axis represents the distance from the axis [m]. The black, magenta, green and*

*yellow dashed lines represent different LWC thresholds for a cloudy pixel (see legend*

*for values). The background represents the condensation (red) and evaporation rate*

*(blue) [g kg$^{-1}$ s$^{-1}$].*

**Specific Comments:**

SC1) line 24 Abstract. The values Bcore, Rcore, Wcore are not defined yet and should be either defined or excluded from the abstract.

SA1) A sentence defining the mentioned values was added to the abstract: "***Three core***

***definitions are examined: positive vertical velocity (Wcore), supersaturation***

***(RHcore), and positive buoyancy (Bcore).***".

SC2) Line 45. The text reads: "detrainment while losing mass". In cloud physics detrainment is "large scale" outflow, typically near cloud top. Interaction of clouds with environment is characterized by entrainment and mixing. The authors supposedly mean that small cloud volumes leaving the parent cloud loss their mass by evaporation.

SA2) Based on our understanding detrainment is the same as entrainment, just opposite wind flow (see AMS glossary). A cloud cannot expand horizontally (which is the case here) by entrainment, only by detrainment where the wind vectors are from in the cloud outwards. After detrainment into the non-cloudy environment, mixing occurs and the final result is either a cloudy (with less LWC) or non-cloudy pixel. We try to clarify the sentence as follows: ***"In clean clouds larger droplets evaporate much slower, enabling preservation of cloud size and even increase by detrainment and dilution (volume increase while losing mass)"***.

SC3) Line 57. what difference of DSD do you mean?

SA3) The differences in DSD mentioned in line 57 are explained in the following line in the text. We have reformulated these few sentences in order to avoid confusion (see also SA4 below): ***"Aerosols act as cloud condensation nuclei (CCN) during heterogeneous nucleation of cloud droplets(Köhler, 1936; Mason and Chien, 1962).The number, size, and composition of aerosol distribution yields differences in the initial cloud droplet size distribution (DSD). Polluted clouds (i.e. more aerosols) have more, but smaller droplets, and a narrower DSD compared to clean clouds (Andreae et al., 2004; Twomey, 1977)."***.

SC4) Line 58. The sentence is not clear or not correct. The nucleation itself that takes place at rN>rNcrit does not accompanied by decrease of S. S decreases as a result of diffusional growth of nucleated droplets.

SA4) Thank you for this comment. We did not intend to say supersaturation is reduced by nucleation, but rather the existence of aerosols enable droplet activation in lower S than in a pristine atmosphere with no aerosols. We have changed this part to be clearer, as seen in SA3 above.

SC5) Line 77. Strictly speaking, the diffusion growth equation is not symmetric with respect of processes of condensation/evaporation. This asymmetry is considered, sometimes, as a mechanism of DSD broadening (e.g. Korolev, 1995, JAS, 52, 3620-

3634).

SA5) Thank you for this comment. Analytically speaking, the diffusion equation should be symmetric for condensation and evaporation, but this is only in theory. Thus, we have removed this sentence from the revised text.

SC6) Line 161. Comment concerning the cloud core definitions. The condition W>0

takes place in cloud cores of devolving clouds. Since the time period of cloud developing is relatively short, effects of mixing with surrounding air may not be significant (depending of cloud size and W). At dissolving stage, W<0. So, there is no cloud core in your definition. At the same time LWC in the cloud may has obvious maximum in the cloud center (interior).

SA6) We note that our choices of cloud core definitions are based on previous works, as listed in Part I: "*Considering convective clouds, there are several objective measures that*

*have been used in previous works for separating a cloud's core from its margins (will*

*be referred to as physical cores hereafter). In deep convective cloud simulations the*

*core is usually defined by the updrafts' magnitude using a certain threshold, usually*

$W>1\ m\cdot s^{-1}$ *(Khairoutdinov et al., 2009; Kumar et al., 2015; Lebo and Seinfeld, 2011;*

*Morrison, 2012). Studies on warm cumulus clouds have defined the clouds' core as*

*parts with positive buoyancy and positive updrafts (Dawe and Austin, 2012; de Roode*

*et al., 2012; Heus and Jonker, 2008; Siebesma and Cuijpers, 1995) or solely regions*

*with positively buoyancy (Heus and Seifert, 2013; Seigel, 2014). More recently, cloud*

*partition to regions of supersaturation and sub-saturation has been used to define*

*the cloud core in single cloud simulations (Dagan et al., 2015).*".

To our knowledge, previous works use a LWC threshold for cloud definition but never for core definition. The case the reviewer describes where LWC has a maximum in the cloud center (or RHcore for example) and there's no Wcore may indeed exist. Due to that, in Part I we define a cloud geometrical core (center of gravity or centroid), and compare its location with the cloud physical core (Wcore, RHcore, Bcore). We quote some of the conclusions here: "*With respect to cloud morphology, the majority of*

*clouds are composed from single cores (for all core types), located near the cloud*

*centroid/COG, and fit the intuitive core-shell model of decreasing core parameter*

*values from cloud center to periphery. This is especially true during cloud growth, as*

*during dissipation the cores may decouple from the geometrical core and often*
*comprise just a few isolated pixels at the cloud's edges.* ".

Regarding the Wcore definition, we quote from Sect 2.3 in Part I: "*We note that setting*
*the core thresholds to positive values (>0) may increase the amount of non-convective*
*pixels which are classified as part of a physical core, especially for the Wcore. Indeed,*
*taking higher thresholds for the Wcore (e.g. $W > 0.2\ ms^{-1}$) decreases the Wcore*
*extent in the cloud and reduces the variance of Wcore fractions between different*
*clouds in a cloud field (as seen in Fig. 4). Nevertheless, any threshold taken is*
*subjective in nature, while the positive vertical velocity definition is process based and*
*objective.*".

Later in that paper, we show that the Wcore is actually much more "well-behaved" than
expected, so that clouds typically have a single Wcore, rather that multiple small
Wcores around the cloud. As is written in the text: "*For the Bcore, RHcore, and*
*Wcore, 68%, 79%, and 81% of the cloud scatter analyzed (which contain a core) have*
*a single core, respectively. Thus, most clouds have a single core. Moreover, it is more*
*probable to find multiple buoyancy cores in a cloud than vertical velocity cores. This*
*is surprising given our choice of "weak" Wcore thresholds (i.e. positive values) and*
*indicates that vertical velocity patterns are relatively well-behaved in cumulus clouds,*
*at least for the LES scales chosen here.*".

SC7) line 184. How can be explained updrafts at B<0? Gravity waves?
SA7) Similar to answer SA6, this issue is also treated in details in Part I. Specifically,
updrafts with negative buoyancy are a very common feature in shallow cumulus fields.
This issue is discussed in depth in Part I, here we quote some of the relevant text: "*…for*
*the adiabatic column case, Bcore is always a proper subset of Wcore (i.e. Bcore $\subset$*
*Wcore. These effects are commonly seen in warm convective cloud fields where*
*permanent vertical layers of negative buoyancy (but with updrafts) within clouds*
*typically exist at the bottom and top regions of the cloudy layer (Betts, 1973; de Roode*
*and Bretherton, 2003; Garstang and Betts, 1974; Grant and Lock, 2004; Heus et al.,*
*2009; Neggers et al., 2007).*", and also: "*The vertical velocity equation dictates that*
*buoyancy is the main production term (de Roode et al., 2012; Romps and Charn,*
*2015), and is balanced by perturbation pressure gradients and mixing (on grid and*
*sub-grid scales). Thus, all changes of magnitude (and sign) in vertical velocity should*

*lag the changes in buoyancy. This is the basis of convective overshooting and*
*cumulus formation in the transition layer*".

SC8) Line 200. In such case it is difficult to call the condition W>0 as definition of
cloud core. Small positive W can take place all over cloud just by turbulent fluctuations.
Katzwinkel et al. (2014) and Schmeissner et al. 2015 determine cloud interior as
LWC>0.2 gm$^{-3}$ this condition guaranties that the region chosen is in the cloud interior.
SA8) We note that the purpose of these papers (including Part I) is to gain the most
general understanding on the partition of cloud to core and margin, using the most
general definitions. As mentioned in MA3, applying a LWC>0.2 gm$^{-3}$ threshold can
exclude much of the cloud, while our goal is to look at the entire cloud. Regarding small
W, as explained in SA6 above, adding random thresholds for a core definition is
unphysical in our opinion, and can be very sensitive to a specific model or case study.
On the other hand, taking positive vs. negative values partitions the cloud based on
purely physical considerations, that can also be applied to other works.  Nevertheless,
as shown in SA6, the occurrence of small positive W pixels is less common than one
would think. Thus, for sake of consistency and generality, once a definition is set for
cloud core, even a small Wcore like the one mentioned in line 200 is considered a core.
.
SC9) line 202. Fig 1. Figure caption. Define LHS axis, RHS axis. What is "other core
types"?
SA9) Thank you, we have changed LHS and RHS to left axis and right axis. The caption
was rephrased as follows: "*Time series of pixel fractions [%] of one core type within*
*another, for the respective simulation types*".

SC10) lines 236-240 The mechanism proposed requires additional justification. The
other option is that in low CCN case drizzle and rain drops rapidly fall down, so LWC
is very low in the subsiding of the air. Another possible mechanism is turbulent mixing
between warm core and colder margin air. This mixing should lead to an increase in T,
i.e to appearance of positive buoyancy. Can you justify the mechanism that is proposed
in the study? What is the cloud stage? Developing or dissolving? Do you see this effect
at cloud center or cloud periphery?
SA10) The mechanism we suggest for the pockets of positive buoyancy has been
thoroughly checked and is a major part of this paper. We start Sect. 3.2 by noting a main point from Part I, that mixing of cloudy and non-cloudy air or core with margin air almost always (except extreme unlikely cases) reduces buoyancy: "*The theoretical*

*arguments in PTI showed that $B_{core}$ should be the smallest of the three. This was*

*shown for both the adiabatic cloud column case and also the non-adiabatic case*

*where entrainment mixing and consequent evaporation has a strong net negative*

*effect has on cloud buoyancy*". The other option that the reviewer is referring to is the depletion of LWC and hence a smaller water loading term in the buoyancy equation, however, this was checked and the water loading term is as large in the positive buoyancy pockets as in their surroundings.  We added this sentence to the text: "*The*

*liquid water content buoyancy term (not shown here) is always negative and typically*

*increases (in absolute value) with increase in vertical velocity or total buoyancy*".

Throughout the paper (see Figs. 5, 7) and also in Part I (see Fig. RA2 below), we find these pockets of positive buoyancy mostly during late mature and dissipation stages of the cloud, after the initial main convective core has disappeared. This effect is mostly attributed to the cloud periphery but can also be in the cloud center for small dissipating clouds. Both sections 3.2 (for single cloud) and 4.2 (for cloud field) provide proof and attempt to justify the assumption that positive buoyancy is formed due to heating in downdrafts during cloud dissipation.

Finally, we have altered the definition of this effect from "adiabatic heating" to "downdraft buoyancy production" and added a more rigorous description of the effect based on previous theoretical and observational works. The description is as follows: "*Although not usually*

*the focus of studies, the existence of positively buoyant downdrafts in convective*

*clouds has been reported in both observations (Igau et al., 1999; Wei et al., 1998) and*

*simulations (Xu and Randall, 2001; Zhao and Austin, 2005a, 2005b). A possible*

*explanation for this can be deduced from previous theoretical studies predicting*

*mixing induced downdrafts in cumulus clouds  (Betts and Silva Dias, 1979; Betts,*

*1982). It was shown that in some cases cloud - environment mixtures are negatively*

*buoyant (while still containing liquid water) and the consequent downdrafts can*

*sometimes descend only part way down to the cloud base before reaching neutral*

*buoyancy. Similar to convective overshooting, parcels with negative vertical*

*momentum may then "undershoot" the downdraft equilibrium level and turn*

*positively buoyant while the downdraft weakens. One can therefore expect the*

*magnitude of positive buoyancy within the downdraft to reach a maximum when the*

*velocity approached zero.  Hereafter we refer to positive buoyancy production within*
*downdrafts as downdraft buoyancy.*

*Downdraft buoyancy production occurs frequently in cumulus fields because the*
*negatively buoyant downdrafts follow a warming lapse rate which is more unstable*
*than the environmental one, which is typically between the dry adiabat and moist*
*adiabat (as is the case for the Hawaiian and BOMEX profiles simulated in this work).*
*On one extreme, a descending parcel is least buoyant (i.e. coolest) when evaporation*
*(after mixing) keeps it just barely saturated (Paluch and Breed, 1984)(also PT1) so*
*that the lapse rate of descent tends to moist adiabatic and may remain negatively*
*buoyant. On the other extreme, if little to no evaporation of liquid water occurs, the*
*descent will follow the dry adiabat and switch to neutral (and then positive) buoyancy*
*rapidly. Thus, the ability of a negatively buoyant cloudy downdraft to sustain itself*
*depends on continuous inflow of liquid water (by mixing) and its consequent*
*evaporation (Knupp and Cotton, 1985). ".*

[Figure]

*Figure RA2. Four vertical cross-sections (at t=8, 20, 30, 40 minutes) during the single*
*cloud simulation. Y-axis represents height [m] and X-axis represents the distance from*
*the axis [m]. The black, magenta, green and yellow lines represent the cloud,*
$W_{core}, RH_{core}$ *and* $B_{core}$, *respectively. The black arrows represent the wind, the*
*background represents the condensation (red) and evaporation rate (blue) [g kg$^{-1}$ s$^{-1}$],*
*and the black asterisks indicate the vertical location of the cloud centroid. Note that in*

*some cases the lines indicating core boundaries overlap (mainly seen for RH and W*
*cores).*

SC11) Line 248 How is cloud margin region defined and calculated?

SA11) Core and margin are defined according to the three definitions used throughout the paper, Wcore, Bcore, and RHcore. We reformulated the sentence to clarify this point: "***Here we evaluate how aerosol effects within the core and margin (using the***

***three core definitions) affect the cloud characteristics***".

SC12) line 254. It is difficult to see the non-monotonic dependence. We see that the maximum cloud mass takes place at high CCN concentration, but lifetime is larger for the low CCN concentration case.

SA12) The non-monotonic dependence referred to in that specific sentence is only that of total cloud mass, as quoted from the text: "***A non-monotonic dependency of total***

***cloud mass on aerosol concentration is seen, showing a maximum for the***

***intermediate concentration. This type of dependency has been previously reported for***

***warm cumulus clouds (Dagan et al., 2015; Savane et al., 2015).***". The effect isn't very large, but in Fig. 3 it is clearly seen that the intermediate concentration reaches the maximum total mass. References for this behavior are given.

SC13) line 285. I do not fully agree with the interpretation. In case of high CCN

concentration droplets are small and mixing with surrounding leads to fast complete evaporation of the droplets. Moreover, small droplets fully and easily evaporate also at

W<0. At the same time, larger droplets formed at low CCN concentration evaporate only partially. Why is it necessary to focus on the weak effect of the differences in the evaporation rates?

SA13) Thank you for this comment. We agree with the reviewer that there are additional parts in the interpretation that should be explained in more details beside the part of differences in diffusion efficiencies (which includes different rates and different droplet surface areas for diffusion process to occur). We now also emphasize the different DSD before evaporation starts which impact the cloud lifetime and cloud volume. Here are a few changes in the text: "***These results with respect to cloud volume***

*can be attributed to the smaller drop sizes and higher diffusion efficiencies with*
*increase in aerosol concentration.*", and: "*The polluted cloud is composed of small*
*drops, evaporates its margin regions efficiently, and is thus limited in dilution growth.*
*The clean cloud is composed of larger drops, less efficient in evaporating its margins,*
*and hence can grow by dilution of its LWC upon a larger volume.*", and: "*The clean*
*cloud shows opposite behavior, with extremes of large super-saturation during cloud*
*growth (initial stages) and large sub-saturation during cloud dissipation (final*
*stages). The large super-saturation can be explained by the low diffusion efficiency,*
*but the large sub-saturation also takes into consideration the larger drop sizes which*
*take more time to evaporate*"

SC14) line 293 At the dissolving stage cloud air descends, i.e. W<0 within cloud body
(Schmeisner et al. 2015). The subsiding dramatically decreases RH and leads to droplet
evaporation. It is natural, that small droplets evaporate first. This decreases the life time
of clouds in polluted air. It would be important to separate two effects: turbulent mixing
of clouds with surrounding and their evaporation at W<0. Note that small Cu often
dissipate and evaporate within the inversion layer, where turbulence (i.e. mixing is
weak). In such case, namely subsiding place dominating role in cloud dissolving.

SA14) The aforementioned line raises the point that once dissipation commences the
only method of cloud volume growth is by dilution via mixing with the environment.
If precipitation below the LCL (lifting condensation level) is excluded, this dilution can
only be attributed to mixing and not subsidence.

SC15) line 295. Which effect? How can precipitation be considered as a method of...?
please reword the sentence. What kind of expansion can be induced by precipitation?
Why the "choice to focus on volume above initial cloud base excludes this effect"? If
the precipitation-induced cooling leads to the formation of new clouds, it is impossible
to exclude the effect by the choice of the altitude, above which cloud properties are
considered.

SA15) Continuing the previous answer (SA14), we wanted to differentiate between
cloud volume expansions due to dilution versus cloud volume expansion due to
precipitation below the cloud base. Since we have significant precipitation in the clean
case, cloud mass descends below the initial cloud base (approximately the LCL) and
increases the cloud volume significantly. For better comparison with the other more polluted cases, we took only the cloudy pixels above the initial cloud base (which is equal for all simulations) and thus volume changes can be attributed to other effects than precipitation below the cloud base.

SC16) line 295 Detrainment is the outflow of cloud mass from the cloud. It cannot change the cloud properties. Entrainment of dry environment air into the cloud indeed can lead to subsaturation. Regular (non-turbulent) entrainment takes place near cloud base. At later cloud edges lateral turbulent entrainment and mixing takes place.
SA16) Please see SA2 on this issue. We define detrainment as the opposite of entrainment (i.e. air flowing out of cloud). The outflow of air from the cloud can then mix with the surrounding similarly to when air is entrained into a cloud and mixes.

SC17) line 298. The effect of dilution depends on cloud width. The larger width the lower the effect of lateral mixing is. The increase or decrease of cloud width depends also on LWC.
SA17) In this section we are comparing three axi-symmetric single clouds which initially have the exact same width. In line 298 we just want to illustrate that the effect of dilution occurs, meaning increase in cloud volume at the same time there is loss of cloud mass: "*A clear indication for dilution is seen in Fig. 3 where between 30 and 35 mins of simulation time both the clean and polluted clouds lose total mass but only the clean cloud increases in total volume*".

SC18) line 300. What is "detrainment growth"?
SA18) The sentence was changed slightly to read: "…*limited in horizontal growth by detrainment* ". As explained above in SA14, SA15, and SA16, growth by detrainment is when clouds may expand in volume after cloudy air is mixed with surrounding environmental air and the droplets do not fully evaporate.

SC19) line 314. To define "diffusion efficiency".
SA19) See MA2 for the issue to defining diffusion efficiency. Specifically, in the mentioned line we removed this term and replaced with "slow diffusion".

SC20) line 327. It is interesting to see the RH (r) profiles in the humid shell around cloud. SA20) Calculating and presenting the RH(r) profiles in the humid shell around all (or a subset of) clouds requires an extensive analysis which is beyond the scope of this work. Moreover, our focus here is on in-cloud processes and cloud properties rather than the effects on the environment adjacent to the cloud. Nevertheless, we note the previous works showing RH(r) have been done, one of them by a member of our research group , showing the distance scale for which RH decreases to the environmental mean.

SC21) line 375. Do you suppose that dissipating clouds may contain dominating cores?

How does it agree with observations?

SA21) As can be seen in Fig. 5, many of the dissipating branch clouds (both larger and smaller ones) can be core dominated, mostly for the Wcore definition but also for a small percentage of clouds using the Bcore definition. We define the dissipation branch according to the COG height so that most dissipating clouds have a cloud base above the LCL and may still be mostly with updrafts. As for observations, according to our knowledge, most observations are biased to larger clouds with cloud base near the LCL

and not the smaller cloud fragment which occupy the cloudy layer. Nevertheless, although not for small cumulus clouds, studies have shown frequent Bcore in downdrafts (Igau et al., 1999; Wei et al., 1998).

SC22) Line 375. Figure 5 shows that simulations with low CCN concentration were performed for the case of 2000 m inversion altitude. Two other simulations were performed for 1500 m altitude. The clouds should be quite different geometrically and microphysically in such cases. How can such clouds be compared?

SA22) Please see MA1 on this issue. The simulations were initialized with the same profile (and same inversion base height) but evolved differently due to different microphysical effects of the clouds on the thermodynamic conditions. In Fig. 5 we present the thermodynamic conditions after 5 h of simulations because we want them to reflect the actual state during the simulation and not the initial state. These different thermodynamic conditions are among the aerosol effects on clouds that are only seen in cloud field simulations.

SC23) Line 417. 1) we see again that there the difference in the inversion level in the simulations. Higher clouds can have larger cloud cover and longer life time etc. 2) It is necessary to add to the figure caption the conditions corresponding to the rows and
columns or refer notations in fig 5.

SA23) For (1), see MA1 and SA22 above explaining the different inversion level
heights. For (2), we have added description to the figure caption: "***CvM space diagrams***
***showing the pixel fractions of Bcore within RHcore (left column), Bcore within***
***Wcore (middle column), and RHcore within Wcore (right column), for the clean (top***
***row), intermediate (middle row), and polluted (bottom row) simulations.***".

SC24) Line 458. I still wonder, how weak downdraft can lead to the temperature of
subsiding air higher than in surrounding. Such subsidence should be actually along the
moist adiabat. Why the downdrafts should be week? It seems that subsidence
accompanied by evaporation leads to cold pool that accelerates formation of new
clouds. It seems that positive buoyancy in the area of weak downdraft is the results of
horizontal mixing between warm zone with W>0 (with high buoyancy) and the cloud
periphery.

SA24) A rigorous explanation for buoyant downdrafts in now added to revised
manuscript (see SA10). Basically, theory shows that cloudy downdrafts follow a lapse
rate more unstable than the environment, meaning that a level of neutral buoyancy is
reached above the cloud base. Since downdrafts have negative vertical momentum
during descent, they will "undershoot" the equilibrium level and become positively
buoyant. We show that this effect is highly dependent on aerosol concentration since
the evaporation rates (and thus determine the lapse rate of descent. As shown in PT1
(and explained in SA10), mixing between positively buoyant and negatively buoyant
regions is unlikely to create positively buoyant mixed parcels.

SC25) line 531. One can suppose that many clouds are isolated even in the clean case.
Why do you illustrate the clean case by merging clouds?

SA25) Section 4.3 in the paper deals with the different relative humidity seen in the
clouds for different aerosol concentrations. As part of this analysis, in Fig. 9 we present
cross-sections of the most massive clouds for each simulation. In line 531 we just
explain to the reader than the most massive clean cloud is actually composed to two
large clouds that merge together and are connected by a few pixels (as can be clearly
seen in Fig. 9). We choose to mention this because this is very typical of the clean case
and a characteristic worth knowing (in our opinion), where precipitation promotes cold pools and later significant merging. We also mention this in the beginning of Sect. 4.1: "***The clean simulation (25 cm$^{-3}$) shows two disconnected regions of cloud scatter: one which is adjacent to the adiabatic approximation and one of mainly small mass and high COG clouds. The former region includes both clouds during their growth stages (smaller masses, LWP < 10 g m$^{-2}$) and large precipitating entities (larger masses, LWP > 10 g m$^{-2}$) which form due to merging processes (Heiblum et al., 2016b).***", and later in that section: "***We note that the higher cloud masses reached by lower aerosol concentration simulation can be explained by cloud field organization effects due to precipitation (i.e. increased merging of clouds) rather than increased cloud condensation (Heiblum et al., 2016b; Seigel, 2014)***.".

SC26) Line 572. So, the formation of low RH is the result of averaging over wider layers which contain the inversion layer and layer below LCL. Please confirm.

SA26) Exactly, the large clouds' margin regions may include areas in the inversion layer and layer below the LCL, and thus we may get low mean margin RH.

SC27) Line 619. Term convection is not suitable. Besides, if you want to compare T with surrounding, moist adiabatic cooling results in heating as compared with the surrounding.

SA27) In the revised manuscript we have changed the terms of the two positive buoyancy production processes to updraft buoyancy production and downdraft buoyancy production.

SC28) Line 621. Adiabatic heating is also not exact term. In the situation considered adiabatic heating is accompanied by turbulent mixing and droplet evaporation. So, many factors determine T in this area, so process is not adiabatic.

SA28) Thank you for this comment, it is true that a cloudy downdraft will likely not descend purely adiabatically. As described in the revised manuscript (and SA10), the descent of a cloudy parcel (during entrainment) will be following a lapse rate somewhere between the moist adiabat and dry adiabat, which represent the two extreme cases. We have removed the term "adiabatic heating" from the revised text.

Oxford, UK

Corresponding Email – ilan.koren@weizmann.ac.il

**Abstract:**

The effects of aerosol on warm convective cloud cores are evaluated using single cloud and cloud field simulations. Three core definitions are examined: positive vertical velocity (Wcore), supersaturation (RHcore), and positive buoyancy (Bcore). As presented in Part I, the property Bcore ⊂ RHcore ⊂ Wcore is seen during growth of warm convective clouds. We show that this property is kept irrespective of aerosol concentration. During dissipation core fractions generally decrease with less overlap between cores. However, for clouds that develop in low aerosol concentrations capable of producing precipitation, Bcore and subsequently Wcore volume fractions may increase during dissipation (i.e. loss of cloud mass). The RHcore volume fraction decreases during cloud lifetime and shows minor sensitivity to aerosol concentration.

It is shown that a Bcore forms due to two processes: i)  Convective updrafts

– condensation within supersaturated updrafts and release of latent heat, ii)

Dissipative downdrafts – sub-saturated cloudy downdrafts that warm during descent "undershoot" the level of neutral buoyancy. The former process occurs during cloud growth for all aerosol concentrations. The latter process only occurs for low aerosol concentrations during dissipation and precipitation stages where large mean drop sizes permit slow evaporation rates and sub-saturation during descent.

The aerosol effect on the diffusion efficiencies play a crucial role in the development of the cloud and its partition to core and margin. Using the RHcore definition, it is shown that the total cloud mass is mostly dictated by core processes, while the total cloud volume is mostly dictated by margin processes. Increase in aerosol concentration increases the core (mass and volume) due to enhanced condensation but also decreases the margin due to evaporation. In clean clouds larger droplets evaporate much slower, enabling preservation of cloud size and even increase by detrainment and dilution (volume increase while losing mass). This explains how despite having smaller cores and less mass, cleaner clouds may live longer and grow to larger sizes.

**1. Introduction**

Aerosols remain one of the largest sources of uncertainty in climate predictions, mainly via their effects on clouds (IPCC, 2013). Here we focus on the aerosol effects on warm clouds. Aerosols act as cloud condensation nuclei (CCN) during heterogeneous nucleation of cloud droplets (Köhler, 1936; Mason and Chien, 1962). The number, size, and composition of aerosol distribution yields differences in the initial cloud droplet size distribution (DSD). Polluted clouds (i.e. more aerosols) have more, but smaller droplets, and a narrower DSD compared to clean clouds (Andreae et al., 2004; Twomey,

1977). Changes in the initial DSD drive various effects and feedbacks on the cloud's evolution and key processes, such as: droplet mobility, condensation/evaporation budgets, collision-coalescence, and entrainment (Jiang et al., 2006; Koren et al., 2015;

Small et al., 2009; Xue and Feingold, 2006).

It is well known that an abundance of small droplets in a cloud (a narrow DSD) reduces the efficiency of the collision-coalescence process (Squires, 1958; Twomey, 1977;

Warner, 1968), prolongs the diffusional growth time (Khain et al., 2005; Wang, 2005), and delays or even completely suppresses the initiation of precipitation (Albrecht, 1989;

Hudson and Mishra, 2007; Hudson and Yum, 2001; L'Ecuyer et al., 2009). Moreover, in-cloud condensational growth is more efficient in consuming supersaturation because of the larger surface area-to-volume ratio of droplets (Dagan et al., 2015a, 2015b;

Mordy, 1959; Pinsky et al., 2013; Reutter et al., 2009; Seiki and Nakajima, 2014).

We note that throughout this work the word *efficient* will be used to describe both the rate and the total change of mass attributed to a microphysical process. The processes described above enable the more polluted cloud to condense more water and intensify its growth via increased release of latent heat (Kogan and Martin, 1994; Koren et al., 2014; Saleeby et al., 2015; Sheffield et al., 2015). The smaller droplets are also pushed higher in the atmosphere due to larger droplet mobility (Koren et al., 2014,
2015).

However, the increase in aerosol amount yields suppressing effects as well.
An opposite effect should take place
in the sub saturated regions of the cloud, where more numerous and smaller droplets
increase the evaporation rate and loss of cloud mass (Grant and van den Heever, 2015;
Saleeby et al., 2015; Storer and van den Heever, 2013). Henceforth evaporation will be
referred to as a process (i.e. change of mass per unit time) rather than complete
evaporation of a water drop. Increased evaporation can promote entrainment mixing
which in turn mixes more sub saturated air into the cloud and further promotes
evaporation (Jiang et al., 2006; Small et al., 2009; Xue and Feingold, 2006), effectively
initiating a positive feedback between evaporation and mixing with the eventual
suppression of cloud growth. This effect may also be accompanied by a suppressing
effect of the larger water loading in polluted clouds which contain more liquid water
mass.

The competition between those opposing processes that are driven by enhanced aerosol
loading determines the net aerosol effect on cloud properties such as cloud fraction,
lifetime, albedo, mass, size, and precipitation amount. However, the sign and magnitude
of such effects are non-trivial (Jiang and Feingold, 2006). Previous studies report
opposing findings regarding the total aerosol effects on warm clouds (Altaratz et al.,
2014). Some studies suggest cloud invigoration by aerosols (bigger and deeper clouds)
(Dey et al., 2011; Kaufman et al., 2005; Koren et al., 2014; Yuan et al., 2011) while
some suggest cloud suppression or no effect at all (Jiang and Feingold, 2006; Li et al.,
2011; Savane et al., 2015; Xue et al., 2008). Moreover, other work has shown that the
precipitation susceptibility (i.e. quantifies the sensitivity of precipitation to the aerosol
increase) has a non-monotonic behavior that reaches its maximum at intermediate LWP
values (Sorooshian et al., 2009), implying that the resultant aerosol effects are heavily
dependent on cloud type and environmental conditions (Khain et al., 2008)

A different
approach to aerosol effects suggests that cloud systems can be buffered to microphysical effects (Stevens and Feingold, 2009). Several studies have shown that
given enough time for the cloud system to reach steady state, cloud macro-physical
parameters (e.g. cloud fraction, rain yield) show similar results for various aerosol
concentrations (Carrió and Cotton, 2014; Glassmeier and Lohmann, 2018; Seifert et al.,
2015). Based on the idea that clouds can be partitioned to aerosol-limited, updraft-
limited, or aerosol and updraft sensitive regimes (Reutter et al., 2009), a unified theory
for the contradicting results regarding aerosol effects was suggested (Dagan et al.,
2015b)was shown in recent work (Dagan et al., 2015b). It was shown that. Given an
aerosol range that covers all three regimes, the competition between opposite processes
leads to an optimum value of aerosol concentration regarding various cloud properties
like total mass, cloud top, or rain (Dagan et al., 2015b). A cloud that develops under
low aerosol concentration is aerosol limited, as it does not have enough collective
droplet surface area to consume the available water vapor. On the other side of the trend,
a cloud that develops in polluted environment (with more aerosols than the optimum)
is influenced significantly by enhanced entrainment and larger water loading, causing
suppression of cloud development. The optimal concentration is a function of the
thermodynamic conditions (temperature and humidity profiles) and cloud size.

[revised manuscript text omitted]

Although not usually the focus of studies, the existence of positively buoyant downdrafts in convective clouds has been reported in both observations (Igau et al., 1999; Wei et al., 1998) and simulations (Xu and Randall, 2001; Zhao and Austin,

2005a, 2005b). A possible explanation for this can be deduced from previous theoretical studies predicting mixing induced downdrafts in cumulus clouds (Betts and Silva Dias,

1979; Betts, 1982). It was shown that in some cases cloud - environment mixtures are negatively buoyant (while still containing liquid water) and the consequent downdrafts can sometimes descend only part way down to the cloud base before reaching neutral buoyancy. Similar to convective overshooting, parcels with negative vertical momentum may then "undershoot" the downdraft equilibrium level and turn positively buoyant while the downdraft weakens. One can therefore expect the magnitude of positive buoyancy within the downdraft to reach a maximum when the velocity approaches zero. Hereafter we refer to positive buoyancy production within downdrafts as downdraft buoyancy.

Downdraft buoyancy production occurs frequently in cumulus fields because the negatively buoyant downdrafts follow a warming lapse rate which is more unstable than the environmental one, which is typically between the dry adiabat and moist adiabat (as is the case for the Hawaiian and BOMEX profiles simulated in this work). On one extreme, a descending parcel is least buoyant (i.e. coolest) when evaporation (after mixing) keeps it just barely saturated (Paluch and Breed, 1984, also PTI) so that the lapse rate of descent tends to moist adiabatic and may remain negatively buoyant. On the other extreme, if little to no evaporation of liquid water occurs, the descent will follow the dry adiabat and switch to neutral (and then positive) buoyancy rapidly. Thus, the ability of a negatively buoyant cloudy downdraft to sustain itself depends on continuous inflow of liquid water (by mixing) and its consequent evaporation (Knupp and Cotton, 1985).

Indeed, the results in Fig. 2 match the hypothesis explained above, where positively buoyant downdrafts are warmer than the environment, and tend to show larger buoyancy values for weaker downdrafts velocities (especially for the intermediate case). Further analysis also shows that the more unsaturated the downdrafts (indicated also by low $B_{Qv}$), the larger the positive buoyancy. Moreover, the occurrence during
precipitating stages and for lower aerosol concentrations indicates that slow
evaporation due to larger
drop sizes is crucial for downdraft
buoyancy production, enabling a near dry adiabatic lapse rate during descent.

[revised manuscript text omitted]

It should be noted that horizontal dashed lines in Fig. 5 represent the inversion base height after 5 hours of simulation (approximately middle of simulation), where an increase in the inversion base height is seen with decrease in aerosol concentration. This is due to increased net warming in the upper cloudy layer (i.e., release of latent heat during condensation with reduced local evaporation) with increase in precipitation (Dagan et al., 2016; Heiblum et al., 2016b), which raises the inversion base.

The results in Fig. 5 show a consistent behavior of the core volume fractions for all aerosol concentrations, where the Wcore type shows the largest fractions and the

Bcore type shows the smallest fractions. The Wcore and RHcore generally show a decrease in core fractions along the growing branch while the Bcore fraction initially increase with cloud growth and then decrease for the large mass growing clouds. The percentages in the panel legends (Fig. 5) indicate the fraction of clouds (out of the scatter) which are core dominated with respect to volume ($f_{vol} > 0.5$). For all concentrations, less than 7% of clouds are Bcore dominated while more than 55% are Wcore dominated (with RHcore percentages somewhere in between). The Bcore typically occupies a small portion of a typical cloud volume while the Wcore typically occupies most of the cloud. The mean cloud area (proportional to scatter point size) shows an increase with increase in mean clouds LWP.

These results are consistent with PTI and the single cloud simulations in Sect. 3.1. Nevertheless, some significant aerosol effects on the partition to core types can be seen. Focusing on the growing branch first (i.e. clouds located near the adiabat), we note the following:

1) For the RHcore type, the core volume fractions of clouds after formation (i.e. with small mass) increase with decreasing aerosol concentration. This effect was also seen for the single cloud simulations and can be explained by the reduced efficiency of super-saturation consumption for fewer aerosols.

2) The Bcore volume fraction increases at smaller mass values (or earlier in cloud's lifetime) and to higher values for increasing aerosol concentration. This effect is complimentary to the previous one, since efficient consumption of super-saturation should result in more latent heat release and positive buoyancy.

3) The core volume fractions of the largest mass clouds increase with increasing aerosol concentration, for all core types.

4) The mean area of large mass clouds increases significantly with decrease in aerosol concentration.

We also note a general increase in the fraction of clouds that are Wcore or RHcore dominated with increase in aerosol concentration. Meaning adding aerosols shifts a cloud from being mostly margin to being mostly core. The Bcore is an exception since the clean case shows the highest fraction of Bcore dominated clouds and both the clean and polluted cases are more Bcore dominated than the intermediate case. This can be explained by the different mechanisms of buoyancy production (see Sect. 3.2 and 4.2), where the polluted case is positively influenced by updraft buoyancy production and a larger core volume fraction while the frequently precipitating clean case is positively influenced by downdraft buoyancy production. For the dissipating branch clouds, a highly variable pattern of core volume fractions can be seen, especially for the small mass clouds. For all aerosol concentrations, these small cloud fragments can be either core dominated, margin dominated, or equally partitioned. One can assume that these differences can be related to the different mechanisms by which cloud fragments form, either by gradual dissipation of a large cloud and by instantaneous shedding of a large cloud. As for aerosol effects on the dissipating clouds, we see the following:

1) Higher RHcore and Wcore volume fractions for gradually dissipating clouds (by rising cloud base) with increase in aerosol concentration. This is manifested by a slower transition from red to blue colors in Fig. 5. It can be explained by the fact that more aerosols increase the convective intensity and extend the core size, while efficiently losing the margins, yielding a higher core volume fraction out of the total cloud.

2) The likelihood to find dissipating cloud fragments with a Bcore increases with decrease in aerosol concentration. For the polluted case most of the dissipating clouds lack a Bcore. This effect was seen in Fig. 1 and explained in Sect. 3.2, showing that  downdrafts promote heating and positive buoyancy in low aerosol concentration cases where evaporation efficiency (and hence cooling) is limited. This effect is checked for the cloud field scale in Sect. 4.2.

As opposed to the single cloud simulations (Sect. 3) where cloud lifetime can be easily defined, in cloud field simulations (especially the cleaner cases) many clouds do not live as individual clouds from formation to dissipation but rather split and merge with other clouds continuously (Heiblum et al., 2016b). Thus, in order to evaluate the lifetime evolution of cores in cloud fields, we focus on the growing branch and use cloud mass [kg] as a proxy for the cloud lifetime during its initial and mature stages. We assume that in the vicinity of the growing branch a larger mass corresponds to a later stage in lifetime.

In Fig. 6 the core mass and volume fractions (using the RH definition) of all growing branch clouds are sorted by mass for the three aerosol concentrations. We note that the higher cloud masses reached by lower aerosol concentration simulation can be explained by cloud field organization effects due to precipitation (i.e. increased merging of clouds) rather than increased cloud condensation (Heiblum et al., 2016b; Seigel, 2014). The clean case starts off with the highest core fractions (both mass and volume) which decrease steadily with increase in mass (or increase in lifetime). For all concentrations, most of the cloud mass is concentrated in the core region. The polluted case shows a slight increase in core mass fractions with increase in mass, while the other two cases show decreases in core mass fractions.

The core volume fractions show lower values than the mass fractions. The clean clouds are margin dominated for most masses, and the polluted clouds are core dominated for all masses. The intermediate case is generally confined to values between the other two cases. Figure 6 can be considered comparable with the lower panels in Fig. 3, but excluding the dissipating part of those time series. The similar findings in both figures indicate the robustness of the aerosol effects on core properties in clouds.

Following the analyses of Sect. 3.1, we next test how aerosol concentration affects the subset properties of one core type within another for all clouds in a field (Fig. 7). We focus only on the typically smaller sized cores (Bcore, RHcore) within larger sized cores. Out of the three permutations, the RHcore inside Wcore shows the lowest sensitivity to aerosol. All three growing branches (for the different aerosol concentrations) consistently show that the RHcore is a subset of Wcore (i.e. ) RHcore $\subset$ Wcore) while the dissipation branches show much lower overlap fraction between the two cores.

Generally, for the dissipating clouds, the lower the mass and the higher the COG, the smaller the overlap. The dissipating branches do include a scatter of small cloud for which RHcore $\subset$ Wcore, comprised of small cloud segments which shed off the main core regions of larger clouds. These findings slightly differ from those of the single cloud simulations that show RHcore $\subset$ Wcore for their entire lifetimes while for cloud fields this property breaks downs during dissipation. This difference highlights the importance of cloud interactions (i.e. splitting, merging) and cloud field air flow patterns (i.e. organized advection, updrafts, and downdrafts) in determining the relationships between core types, enabling supersaturation and downdrafts to coincide in small dissipating clouds.

The other two permutations (i.e.  Bcore inside )RHcore, Wcore) show significant changes due to aerosol. For the polluted case, Bcore ⊂ Wcore for nearly all clouds, including clouds at initial stages of dissipation. Similar results are seen for Bcore inside RHcore, but with slightly lower pixel fractions. The polluted case thus illustrates the case of buoyancy production due to convective updraft. For the lower aerosol concentrations, two main aerosol effects are seen:

1) The lower the concentration, the lower the chance that Bcore is a proper subset of the other cores for large growing branch clouds.

2) The lower the concentration, the more prevalent the independent dissipating branch Bcore that has little to no overlap with the other cores.

For the case of Bcore within  RHcore, the lower concentrations show an almost binary scenario where either Bcore ⊂ RHcore or Bcore ∉ RHcore. These result bear similarity with the single cloud simulations, where a quick transition (in time) from Bcore ⊂ RHcore to Bcore ∉ RHcore was seen. These results imply the existence of two different buoyancy production processes ( more in Sect. 4.2), one associated with supersaturation and the other with sub-saturation. In contrary,  pixels fractions of Bcore inside Wcore span the entire range of values (i.e. partial overlaps between the core types), as seen for both single clouds and cloud fields during dissipation. This is to be expected due to the a more direct physical link and feedbacks between the Bcore and Wcore.

**4.2.      Analysis of cloud field buoyancy**

In Sect. 3.2 it was seen that for single clouds, positive buoyancy results from two main mechanisms: i)  convective updrafts - where updrafts promote supersaturation and latent heat release, and thus always positive $B_{Qv}$ and frequently positive $B_T$ , and ii) dissipative downdrafts – where sub-saturated cloudy downdrafts promote a positive $B_T$ and neutral $B_{Qv}$. The latter case is dependent on low evaporation efficiency and hence seen mostly for precipitating stages of low aerosol concentration simulations.

In Fig. 8 we perform a similar test for the cloud field scale. Instead of analyzing pixel by pixel, we check whether each buoyancy core within a cloud is $B_T$ or $B_{Qv}$ dominated.

To quantify this we use a normalized buoyancy dominance parameter

$\frac{\Sigma pixel_{B_T>0} - \Sigma pixel_{B_{Qv}>0}}{\Sigma pixel_{B>0}}$, where a core comprised of only $B_T$  $\geq 0$ ($B_{Qv}$  $\geq 0$) pixels yields

1 (-1). Hence, we expect negative (positive) values to indicate dominance of updraft buoyancy (downdrafts buoyancy).

Analysis of the buoyancy components in the CvM space (right column, Fig. 8) shows that the large majority of clouds are $B_{Qv}$ dominated. For all concentrations, clouds initiate with all pixels showing $B_{Qv}$  $\geq 0$. As clouds develop along the growing branch the Bcore becomes more abundant with $B_T$  $\geq 0$ pixels. This is expected with increasing release of latent heat during cloud growth. During dissipation $B_{Qv}$ again becomes the dominant component for the majority of clouds. The polluted simulation shows an extreme case where all buoyancy cores in the simulation are $B_{Qv}$ dominated, while for the lower concentrations a portion of the dissipating and precipitating clouds are $B_T$ dominated.

Thus, we hypothesize that the polluted simulation only permits buoyancy cores of the updraft type which intersect with the other core types (i.e.

Bcore ∈ RHcore, Wcore), while the lower concentrations also permit buoyancy cores of the downdraft type which do not intersect with the other core types (i.e. Bcore ∉ RHcore,

Wcore). We test this by observing the relation of cloud maximum absolute vertical velocity (left column, Fig. 8) and mean drop size (middle column, Fig. 8)

with the relative dominance of the buoyancy terms. Absolute vertical velocity is chosen to represent both updrafts and downdrafts. The data is further separated to independent (Bcore ∉ RHcore, Wcore) and dependent (

Bcore ∈ RHcore, Wcore) buoyancy subsets of the data, by that separating to buoyant cores within updrafts and downdrafts. Clear aerosol effects are seen on cloud mean drop size and maximal |W|. As expected, there is a decrease in drop size with increase in aerosol concentration and increase in maximal velocity.

Regarding cloud field buoyancy, as predicted the independent buoyancy cores are more
frequently $B_T$ dominated than the dependent buoyancy cores.

The polluted case is populated with dependent cores (white scatter) and shows a classic
pre-precipitation convective growth scenario, where relative dominance of the $B_T$
term increases linearly with increase in cloud mean drop size. A logarithmic
dependence of $B_T$ dominance on maximal |W| is seen, which saturates at high
maximal |W~.~|. This can be explained by the fact increased convection mainly increases
the abundance of pixels with $B_T > 0$, but without changing the fact that the entire
cloud is $B_{Qv} > 0$, so that $B_T$ is unlikely to become the dominant term.

The lower concentrations show a more complex scenario. These simulations show a
superposition of dependent core convective growth behavior (i.e. the scatter pattern
seen for the polluted case) and additional populations of both dependent (other white
scatter points) and independent (black scatter) cores.

The independent cores span all the range of possibilities of $B_T$ and $B_{Qv}$ relative
dominances. They tend to have larger cloud mean drop sizes, and near zero maximum
|W|, indicating that they only form at late non-convective stages of cloud development.
Furthermore, a trend is seen for the subset of scatter that is $B_T$ dominated, where a
positive (negative) correlation between mean drop size (maximal |W|) and $B_T$
dominance is seen. This again stresses the importance of drop size on the formation of
positive buoyancy within downdrafts, and highlights the fact that $B_T$ should be largest
(and most abundant) below the downdraft equilibrium level, when the |W| approaches
zero. The independent cores that are $B_T$ dominated thus fulfill the characteristics of
downdraft buoyancy production process, while the independent cores that are $B_{Qv}$
dominated may originate from larger clouds (shedding mechanism) with high humidity
content, have weak |W|, and are slow to evaporate.

The intermediate simulation shows an additional scatter area of dependent core clouds with increasing of $B_T$$B_T$ relative dominance for lower maximal $|W_⟂|$, located between the independent core clouds and the convective growth core clouds. These clouds may represent a gradual transition from $B_{Qv}$ $B_{Qv}$ dominance to $B_T$$B_T$ dominance during dissipation which is only possible in the intermediate simulation. This scatter area is absent from the clean and polluted simulation. In the former case due to absence of the gradual dissipation pathway, and in the latter case due to efficient evaporation eliminating $B_{core}$ during dissipation. $B_{core}$ during dissipation. We note that the intermediate case shows a slightly higher percentage of clouds that are $B_T$ dominated (see legends in Fig. 8) than the clean case. This can be due to stronger convection in this simulation (i.e. increased $|W|$ range), which favors increased mixing with the dry environment (see Fig. 9) and the formation of unsaturated strong downdrafts that descend below the level of neutral buoyancy.

**4.3.    Aerosol effects on cloud relative humidity**

[revised manuscript text omitted]

**Summary**

In this work we explored how the aerosol effects on warm convective clouds are reflected in their partition to core and margin regions. Following part I of this work (PTI), we evaluated three types of core definitions: positive buoyancy ($B_{core}$),Bcore), super-saturation ($RH_{core}$),RHcore), and positive vertical velocity ($W_{core}$).Wcore). Both single cloud and cloud field models have been used.

For all aerosol concentrations, (clean, intermediate, and polluted) it is shown that the self-contained property of different core types (i.e. $B_{core} \subseteq RH_{core} \subseteq W_{core}$Bcore $\subset$ RHcore $\subset$ Wcore) is maintained for clouds during their growing and mature stages. This is especially robust for the $RH_{core} \subseteq W_{core}$RHcore $\subset$ Wcore subset. The

$W_{core}$Wcore and $RH_{core}$RHcore volume fractions decrease monotonically during cloud growth, while $B_{core}$Bcore initially increases and then decreases after convection ceases. During growth, the $RH_{core}$ ($B_{core}$)RHcore (Bcore) volume fractions are largest for clean (polluted) clouds. This is due to low (high) diffusion efficiencies, respectively, where efficient condensation promotes $B_{core}$Bcore at the expense of theRH$_{core}$.the RHcore.

During dissipation stages cores frequently cease to be subsets of one another and may either increase or decrease in their volume fractions. In cloud fields we also observe small cloud fragments which shed off larger cloud entities. This shedding increases for the lower concentration simulation which produce long-lived large cloud entities. due to cloud merging. These fragments show large variance in volume fraction (for all core types) magnitudes without any consistent behavior. This is due to the fact that they shed off various locations of the cloud. The polluted, non-precipitating cases, are unique in that can one expect the $B_{core}$Bcore to decrease monotonically and remain the smallest and a proper subset of the other cores.

For low aerosol concentration, clouds which are capable of producing precipitationconcentrations, a $B_{core}$Bcore may form during dissipation and exist independently of the other core types. These cores are typically located at the periphery of large clouds, or throughout small precipitation or dissipating cloud fragments. The increase in $B_{core}$Bcore during dissipation typically coincides with large drop sizes and precipitation production. The fluctuations in $B_{core}$Bcore for low concentrations may also create a subsequent $W_{core}$,Wcore, but not of sufficient strength to also create a $RH_{core}$.RHcore. Hence, the $RH_{core}$RHcore can be considered the most "well-behaved" and indicative of cloud lifetime, generally monotonically decreasing in volume fraction irrespective of aerosol concentration.

We show that the $B_{core}$Bcore in the warm convective cases considered here may form by two main processes:

1. ConvectionConvective updrafts: adiabatic cooling within updrafts promotes supersaturation, condensation, and release of latent heat. These cores are characterized by both positive temperature ($B_T > 0$) and humidity ($B_{Qv} > 0$) buoyancy terms.

2. Dissipative downdrafts

2.  sub-saturated cloudy downdrafts follow a lapse rate which is unstable relative to the environmental one. These downdrafts undershoot the equilibrium level and become positively buoyant. These cores are characterized by  positive humidity term ($B_{Qv}$),  positive temperature term ($B_T$)  ($B_T > 0$) but neutral humidity ($B_{Qv} \sim 0$) buoyancy terms.

The updraft buoyancy type is seen for all aerosol concentrations, while the dissipation buoyancy type is only seen for lower aerosol concentrations

. The fact that the downdraft Bcore is absent from polluted clouds highlights the importance of  drop size and its effect on evaporation rate. The high (low) diffusion (collision coalescence) efficiencies in polluted clouds maintain a small mean drop size and enable rapid evaporation during entrainment, causing a  negative effect on buoyancy. For lower concentrations, clouds with a downdraft Bcore only exist during late mature, dissipation, and precipitating stages after drop size has grown considerably. The larger mean drop sizes reduce evaporation rates and the cloudy downdrafts may thus descend nearly dry adiabatically and become positively buoyant.

Focusing on cores using the RH definition, a cloud's mass (volume) is dependent primarily on the processes in its core (margin). The core increases cloud mass by condensation while the margin increases the cloud's volume by mixing with the environment, or dilution. The magnitude of the effects in each region of the cloud is strongly dependent on the aerosol concentration. ~~Increasing the aerosol concentration increases the vapor diffusion rate, minimizing both the super saturation and sub-saturation (absolute) values in the cloud. Thus, polluted clouds are efficient in accumulating water mass but also in losing it. This competition between the core mass gain and margin mass loss regions is what brings about the concept of an optimal aerosol concentration (Dagan et al., 2015b), and explains why more polluted clouds are not necessarily more massive.~~

Polluted clouds are core dominated both in terms of mass and volume, since they can hardly maintain their margins. Clean clouds are also core dominated in terms of mass, but to a lesser degree. Clean clouds tend to be margin dominated in terms of volume for most their lifetimes. Thus, despite weaker convection in the clean clouds, their large, slow evaporating margins enable their cores (and the entire cloud) to exist for longer time spans by applying a large "protecting shield" around the core.

The different diffusion efficiencies are demonstrated by observing the relative humidity (RH) values in clouds. Cleaner clouds show larger variance in RH values. During their growing stages large super-saturation in the core and sub-saturation in the margin can be seen. During their dissipation stages clouds may exist for minutes without any cloud core, with the entire cloud at sub-saturation. Polluted clouds show the opposite, with RH values nearing 100% throughout the cloud, at all stages. Hence, above a certain aerosol concentration, the saturation adjustment approximation (i.e. instant condensation of all super-saturation) can be considered valid. However, the transition from clean to polluted is not always linear. For example, for the largest clouds in the intermediate case have lower margin RH value than both the clean and polluted cases. This is due to the fact that the intermediate case manages to develop taller (than the clean case) clouds with stronger updrafts and downdrafts which entrain drier air from above the inversion layer base, but at the same time is less efficient in evaporating (than the polluted case) water and adjusting the RH to 100%.

**Author Contributions**

RH ran cloud field simulations and conducted the analyses, and wrote the final draft of
paper. LP participated in writing the first draft, and performed single cloud simulations
and relevant analyses. OA, GD, and IK participated in paper editing and discussions.

**Acknowledgements**
The research leading to these results was supported by the Ministry of Science &
Technology, Israel (grant no. 3-14444).

**Figures**

[Figure]

[Figure]

*Figure 1. Left: Time series of core volume fractions (($f_{vol}$ [%], left axis) and surface rain-rate ( [mm hr$^{-1}$], right axis) for the clean (top panel), intermediate (middle panel), and polluted (bottom panel) single cloud simulations. Right: Time series of  pixel fractions ($f_{pixel}$ [%]) of one core type within another, for the respective simulation types. Core volume and pixel fractions are indicated by different line colors (see legends).*

[Figure]

[Figure]

*Figure 2. Scatter plots of pixel total buoyancy [m s⁻²] vs. pixel vertical velocity [m s⁻¹], for the clean (left), intermediate (middle), and polluted (right) simulations. Data includes all cloudy pixels during all time steps. Colors represent magnitude of buoyancy temperature term ($B_T$, upper row) and humidity term ($B_{Qv}$, lower row), where red (blue) shades indicate positive (negative) values. Markers with black dots superimposed represent temporal stages with non-zero surface precipitation. White arrows indicate outlier scatter of pixels with positive buoyancy and negative vertical velocity.*

[Figure]

[Figure]

*Figure 3. Time series of cloud mass ([kg], left column) and cloud volume ([km³], right*
*column) for the different aerosol concentrations simulations (see legend). The total,*
*core, margin, and relative fraction values are shown for each parameter, as indicated*
*by panel titles. The core here is defined according to RH>100% definition.*

[Figure]

[Figure]

*Figure 4. Four temporal snapshots (see panel titles for times) of RH [%] horizontal cross-sections. Panels include the results of different aerosol concentrations (see legend). Cross-sections are obtained by taking the mean RH of all vertical levels for each horizontal distance from the cloud center axis.*

[Figure]

[Figure]

*Figure 5. CvM phase space diagrams of Bcore (left column), RHcore (middle column), and Wcore (right column) volume fractions (f_vol) for all clouds between 3 h and 8 h in the BOMEX simulations. The upper, middle, and lower rows correspond to the clean, intermediate, and polluted aerosol cases. -The red (blue) colors indicate a core f_vol above (below) 0.5. The size of each point in the scatter is proportional to the cloud mean area, where the smallest (largest) point corresponds to an area of 0.01 (11.4) km². The percentage of clouds*

*that are core dominated ($f_{vol} > 0.5$) is included in panel legends.* For an in-depth description of CvM space characteristics, the reader is referred

*to Sect. 2.4 in PTI.*

[Figure]

[Figure]

*Figure 6. Average core mass fraction (left) and volume fraction (right) values for*

*different aerosol concentrations, as indicated in the legend. The average only includes*

*growing branch clouds from within the CvM space (i.e. clouds located in proximity to*

*the adiabat). The core here is defined according to RH>100% definition.*

[Figure]

[Figure]

*Figure 7. CvM space diagrams showing the pixel fractions ($f_{pixel}$) of Bcore within RHcore (left column), Bcore within Wcore (middle column), and RHcore within Wcore (right column), for the clean (top row), intermediate (middle row), and polluted (bottom row) simulations. Bright colors indicate high pixel fractions (large overlap between two core types) while dark colors indicate low pixel fraction (little overlap between two core types). The differences in the scatter density and location for different panels are due to the fact that only clouds which contain a core fraction above zero (for the core in question) are considered.*

[Figure]

[Figure]

*Figure 8. Analysis of dominant buoyancy term within  Bcore of clouds (see text for details). As seen in previous figures, rows represent clean (top), intermediate (middle), and polluted (bottom) simulations. Left column: dependence on maximum absolute vertical velocity within cloud. Middle column: dependence on partition of total cloud mass to cloud droplets and rain drops. Right column: CvM space diagrams of all clouds with  Bcore, where red (blue) shades indicate temperature (humidity) buoyancy terms dominate the cloud. Legends include percentage of clouds that are $B_T$ or $B_{Qv}$ dominated (see text for explanation).*

[Figure]

[Figure]

Figure 9. Left column – Relative Humidity (RH [%]) vertical cross-sections slicing through the center of gravity of the most massive cloud in each simulation. Middle and right columns display CvM space diagrams of mean cloud margin RH and mean cloud core RH, respectively, using the RHcore definition. The upper, middle, and lower panels correspond to the clean, intermediate, and polluted aerosol cases (see panel titles). Notice the different color bar ranges for margin and core mean RH panels.